# A spatially resolved atlas of the human lung characterizes a gland-associated immune niche

Elo Madissoon[1,2,11], Amanda J. Oliver [1,11], Vitalii Kleshchevnikov[1], Anna Wilbrey-Clark[1], Krzysztof Polanski [1], Nathan Richoz[3], Ana Ribeiro Orsi [1,4], Lira Mamanova [1], Liam Bolt[1], Rasa Elmentaite [1], J. Patrick Pett [1], Ni Huang[1], Chuan Xu[1], Peng He [1,2], Monika Dabrowska[1], Sophie Pritchard[1], Liz Tuck [1], Elena Prigmore [1], Shani Perera[1], Andrew Knights[1], Agnes Oszlanczi[1], Adam Hunter[1], Sara F. Vieira [1], Minal Patel[1], Rik G. H. Lindeboom [1], Lia S. Campos[1], Kazuhiko Matsuo[5], Takashi Nakayama [5], Masahiro Yoshida [6], Kaylee B. Worlock [6], Marko Z. Nikolić [6], Nikitas Georgakopoulos [7], Krishnaa T. Mahbubani [7], Kourosh Saeb-Parsy [7], Omer Ali Bayraktar [1], Menna R. Clatworthy[1,3], Oliver Stegle[1,8,9], Natsuhiko Kumasaka [1], Sarah A. Teichmann [1,10] ✉ & Kerstin B. Meyer [1] ✉

Single-cell transcriptomics has allowed unprecedented resolution of cell types/states in the human lung, but their spatial context is less well defined. To (re)define tissue architecture of lung and airways, we profiled five proximal-to-distal locations of healthy human lungs in depth using multi-omic single cell/nuclei and spatial transcriptomics (queryable at lungcellatlas.org). Using computational data integration and analysis, we extend beyond the suspension cell paradigm and discover macro and micro-anatomical tissue compartments including previously unannotated cell types in the epithelial, vascular, stromal and nerve bundle micro-environments. We identify and implicate peribronchial fibroblasts in lung disease. Importantly, we discover and validate a survival niche for IgA plasma cells in the airway submucosal glands (SMG). We show that gland epithelial cells recruit B cells and IgA plasma cells, and promote longevity and antibody secretion locally through expression of CCL28, APRIL and IL-6. This new 'gland-associated immune niche' has implications for respiratory health.

A comprehensive understanding of cells and micro-environments that define lung function is important for reducing the impact of lung diseases, which currently rank third for mortality causes worldwide[1]. In addition to its main role in gas exchange, the lung has an important barrier function. While other mucosal barrier tissues orchestrate adaptive immunity through well-defined mucosa-associated lymphoid tissue (MALT), such secondary lymphoid structures have not been reported in the healthy human lung[2]. The LungMAP and human lung cell atlas (HLCA) consortia[3,4] have harnessed recent advances in single-cell and single-nucleus RNA sequencing (scRNA-seq and snRNA-seq)[5] and

generated a number of atlases characterizing lung cell types across species, health and disease[6–10,11].

Current atlases have prioritized parenchyma tissue, with few studies examining the full depth of the airways. Here we carried out deep tissue profiling from deceased organ donors' healthy lungs and airways, allowing characterization of cell types along the proximal to distal axis of the respiratory tree. We use unbiased spatial transcriptomics (ST) approach to contextualize cell types and states within tissue micro-environments in the healthy human lung and airways, adding a key dimension to the HLCA. In total, we sequenced 129,340 single cells and 63,768 single nuclei and performed Visium ST on 20 tissue sections from human trachea, bronchi, and upper and lower parenchyma. These data and CellTypist automated annotation models are available at lungcellatlas.org as a resource for data download, suspension and spatial gene expression analysis, as well as automated annotation of new datasets. Overall, we distinguished 80 cell types and states, including 11 populations not annotated in previous lung atlas studies. Many of these populations express disease-associated genes highlighted by functional genome-wide association studies (fGWAS) analysis. Our in-depth tissue profiling coupled with spatial genomics reconstructs known tissue micro-environments in the lungs and airways at full molecular breadth. Going beyond known units of cellular organization, we identify a previously undefined immune niche for IgA plasma cells at the airway submucosal glands (SMG).

## Results

### A spatial, multi-omics atlas of human lung and airways

We applied scRNA-seq and snRNA-seq, VDJ-seq and ST to deep tissue samples from five locations across the human lung and airway (Fig. 1a and Supplementary Tables 1–3 and 9) to capture structures such as cartilage, muscle and the SMG (Extended Data Fig. 1a; 'Methods').

In total, 193,108 cells and nuclei were annotated into broad cell type groups as follows: epithelial, immune, erythroid, endothelial and stromal cells. Cells were annotated according to consensus marker genes and naming from other lung studies including the integrated HLCA[10] and LungMAP[12] (Fig. 1b, Extended Data Fig. 1b and Supplementary Table 4). Using Visium ST on 20 tissue sections from five locations and the cell2location[13] algorithm (Supplementary Fig. 1), we assessed the spatial distribution of cells in distinct tissue micro-environments (Supplementary Table 5). Essentially, the tool determines which cell types from suspension data in which abundance could explain the mRNA counts observed in the Visium data. As expected, well-described cell types mapped to their known locations such as ciliated epithelial cells to the lumen of the airway surrounded by basal cells and alveolar type 1 (AT1) and 2 (AT2) cells to lung parenchyma (Fig. 1c,d). To examine differential cell composition related to the sampling locations, donors and protocols used, we used a Poisson linear mixed model to identify the contribution of each technical variable (Methods; Fig. 1e and Supplementary Fig. 2). Different dissociation protocols enriched for specific cell type groups but had little effect (less than 1% of the total variance) on gene expression (Extended Data Fig. 1c).

Highlighting our comprehensive approach, we transcriptionally defined chondrocytes in human lungs (ACAN, CHAD, COL9A3, HAPLN1 and CYTL1; Extended Data Fig. 1d,e) and mapped them to airway cartilage (Fig. 1c,d). Chondrocytes were mostly released using single nuclei sequencing from trachea (Fig. 1e, Extended Data Fig. 1e,f and Supplementary Table 7) and were not present at all in the integrated HLCA[10], demonstrating the utility of our multi-omics, multilocation human lung atlas.

### Rare fibroblasts with immune recruiting properties.

The sequential clustering of fibroblasts identified 11 distinct fibroblast clusters (Fig. 2a,b and Extended Data Fig. 2a,b). We annotated previously described myofibroblasts, mesothelial, adventitial and alveolar fibroblasts[11] (Extended Data Fig. 2c–e and Supplementary Table 4) and seven new subsets, including a rare cell type, termed immune recruiting fibroblasts (IR-fibro). IR-fibro cells expressed the chemokines CCL19 and CCL21 and other marker genes of fibroblast reticular cells and follicular dendritic cells (fDC), which together are responsible for T and B cell positioning in secondary lymphoid organs (Fig. 2c)[14–16]. These cells were mapped to rare immune infiltrates in the bronchus with ST and were validated by multiplexed single molecule FISH (smFISH) (Fig. 2d and Extended Data Fig. 3a, b). The amount of immune infiltrates present in our healthy donors was consistent with a previous study[17]. The gene signature of germinal center fibroblasts from Peyer's Patches[18] also mapped to the immune infiltrate captured by Visium ST, further supporting the similarity of IR-fibros to lymphoid organ stromal cells (Extended Data Fig. 3b). In conclusion, we describe an IR-fibro population with a likely role in immune cell recruitment. Using its newly defined marker genes, this population can also be detected in the HLCA[10].

### Peribronchial and perichondrial fibroblasts.

Two fibroblast populations, both enriched in the airways, were annotated based on their specific mapping around the airway epithelium (peribronchial fibroblasts−PB-fibro) and the cartilage (perichondrial fibroblasts−PC-fibro) (Fig. 2b,e,g and Extended Data Fig. 2b). We uncover the transcriptome for human PB-fibro, consistent with the key protein markers COL15A1 and ENTPD1 (Extended Data Fig. 4a,b). fGWAS analysis, which quantifies systematic associations between cell-type-specific genes and disease-associated SNPs[18], revealed that PB-fibros are linked to lung function measured by FEV1/FVC ratio, the decrease of which is associated with lung diseases such as chronic obstructive pulmonary disease (COPD) (Fig. 2f and Supplementary Table 8). We annotated PB-fibros in a single-cell dataset of COPD and idiopathic pulmonary fibrosis (IPF) patients[8,10] and found a number of COPD-upregulated genes that had previously been associated with lung function (FEV1/FVC) and emphysema/COPD (Extended Data Fig. 4c and Supplementary Table 10)[19–21]. In addition, we found that PB-fibros were reproducibly enriched in IPF patients compared to healthy controls in the Adams et al. (ref. [8]) dataset and the HLCA[10], further implicating them as a key cell type in lung disease (Extended Data Fig. 4d,e).

PC-fibros, like chondrocytes, were enriched in single nuclei data (Fig. 2b). We found analogous marker expression of COL12A1 around the cartilage in the Human Protein Atlas (HPA) (Extended Data Fig. 4f), supporting cell2location mapping of PC-fibro (Fig. 2g). We identified bone development genes LRG4/6 (ref. [22]) along with fibroblast markers in PC-fibros, placing these as an intermediate cell type in a trajectory from adventitial fibroblasts to chondrocytes (Extended Data Fig. 4g,h). PC-fibros express genes causing skeletal abnormalities in humans (Extended Data Fig. 4i), including FLNB and FGFR2, suggesting the relevance of PC-fibros in supporting cartilage functions and related abnormalities[23,24].

### Four distinct cell types in airway peripheral nerves.

Finally, we identified the following four new clusters relating to airway peripheral nerves: myelinating Schwann cells (mSchwann) (NFASC, NCMAP, MBP and PRX), nonmyelinating Schwann cells (nmSchwann) (NGFR, SCN7A, CHD2, L1CAM and NCAM1)[25–28], endoneurial nerve-associated fibroblasts (NAF) (SOX9 and OSR2)[25] and perineurial NAF (SLC2A1 and ITGA6)[25,29] (Extended Data Fig. 5a). nmSchwann and mSchwann cell marker genes were enriched in cell adhesion and myelination gene sets, respectively (Extended Data Fig. 5b,c), with EVX1, a key gene in spinal cord development, identified as a potential regulator of mSchwann cells in the airways. Both mSchwann and nmSchwann expressed peripheral nervous system disease genes (Extended Data Fig. 5d). Localization of these populations in peripheral nerves was validated with bulk RNA-seq across tissues (Extended Data Fig. 5e), Visium ST (Extended Data Fig. 5f), protein staining (Extended Data Fig. 5g–i) and smFISH (Fig. 2i and Extended Data Fig. 5j). We show perineurial NAFs

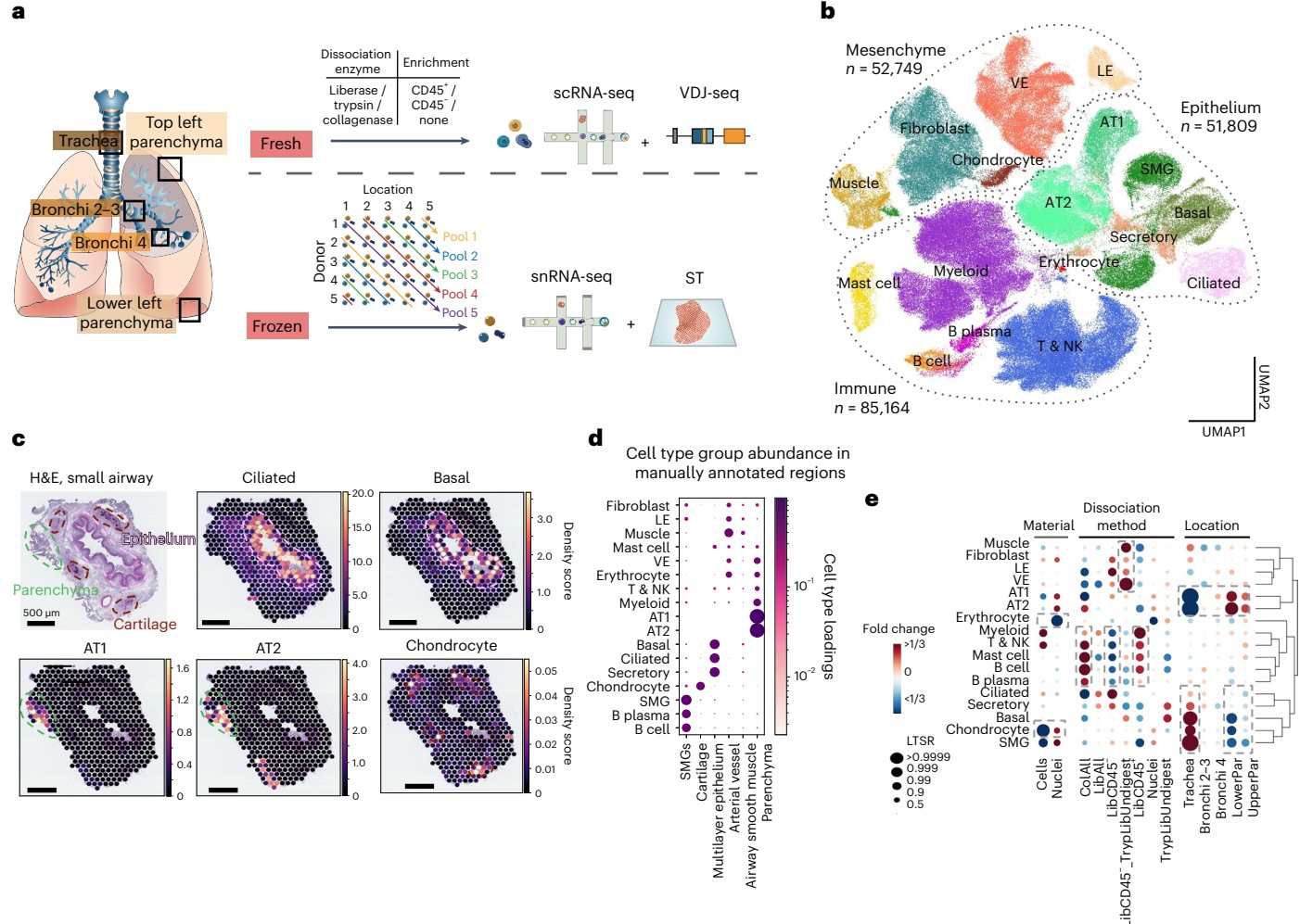

**Fig. 1 | Spatial multi-omics atlas of the human lung allows the identification of cell types and their location. a**, Multi-omics spatial lung atlas experimental design included fresh and frozen sampling from five locations for scRNA-seq (seven donors), sc VDJ-seq (four donors), snRNA-seq (seven donors) and Visium ST (seven donors). Five donors (D) from the frozen samples were pooled into five reactions, each containing different locations (Loc) from donors. **b**, UMAP of all scRNA-seq/snRNA-seq from 193,108 cells/nuclei in total from ten donors. Cells from all major subsets were captured. **c**, cell2location mapping on Visium ST from a bronchi section shows matching of cell types to expected structures. H&E staining and cell abundance estimated by cell2location (density score) for ciliated, basal epithelium, AT1, AT2 and chondrocyte cell types with histology image in the background. Dotted lines circle the epithelium (pink), parenchyma

(green) and cartilage (brown). **d**, Cell type groups are enriched in expected micro-anatomical tissue environments on Visium ST across sections from five donors. Cell type loadings are represented by both dot size and color for cell types annotated in **b** across the manually annotated micro-environments in the Visium data. **e**, Cell type capture is affected by protocol and location. Cell type proportion analysis with fold changes and LTSR score for all cell type groups with regard to the material, protocol and location. Dashed boxes highlight the greatest changes. AT1, alveolar type 1; AT2, alveolar type 2; LE, lymphatic endothelium; VE, vascular endothelium. The number of cells in each cell type group is shown in Supplementary Table 7 and online: https://www.lungcellatlas.org as variable Celltypes_master_high.

surrounding and endoneurial NAFs alongside Schwann cells within the nerve bundle in human airway samples. In conclusion, we have detected and mapped rare stromal cells of airway peripheral nerves.

**Vascular cell types in systemic and pulmonary circulation.** Focusing on vasculature, we could distinguish clusters of pulmonary and systemic circulation by the specific enrichment in parenchyma (pulmonary vasculature, where gas exchange occurs) and trachea (systemic vasculature, providing oxygen to the tissue) (Fig. 3a,b). We also distinguished further cell types in distinct tissue locations using ST: endothelial arterial cells (systemic E-Art-syst and pulmonary E-Art-pulm), smooth muscle cells (non-vascular airway (ASM), pulmonary (SM-pulm) and systemic (SM-syst)) and pericytes (pulmonary (Peri-pulm), systemic (Peri-syst) and venous immune recruiting (IR-Ven-Peri)[11]; Fig. 3a-f, Extended data fig. 6b). Using ST in different

tissue locations, we distinguished further endothelial arterial cell types (systemic arterial endothelia (E-Art-syst) and pulmonary arterial endothelia (E-Art-pulm)) (Fig. 3a,c and Extended Data Fig. 6b), nonvascular airway smooth muscle (ASM) cells, and both pulmonary and systemic smooth muscle/perivascular cells (pulmonary smooth muscle (SM-pulm) and pulmonary pericyte (Peri-pulm), systemic arterial smooth muscle (SM-Art-syst), systemic pericyte (Peri-syst) and venous perivascular cells, that is, immune recruiting perivascular cells (IR-Ven-Peri)[11] (Fig. 3a-c,f). ASM cells lined the airways (Extended Data Fig. 6c), were mainly captured in single nuclei data (Fig. 3d) and had marker genes aligned across tissues with smooth muscle in the HPA and GTEx (Extended Data Fig. 6d, e).

The IR-Ven-Peri expressed *ABCC9* and *ICAM1* but not *CSPG4*, similar to postcapillary venous perivascular cells important for immune cell homing to peripheral lymph nodes[30,31] (Fig. 3d,e). IR-Ven-Peri expressed

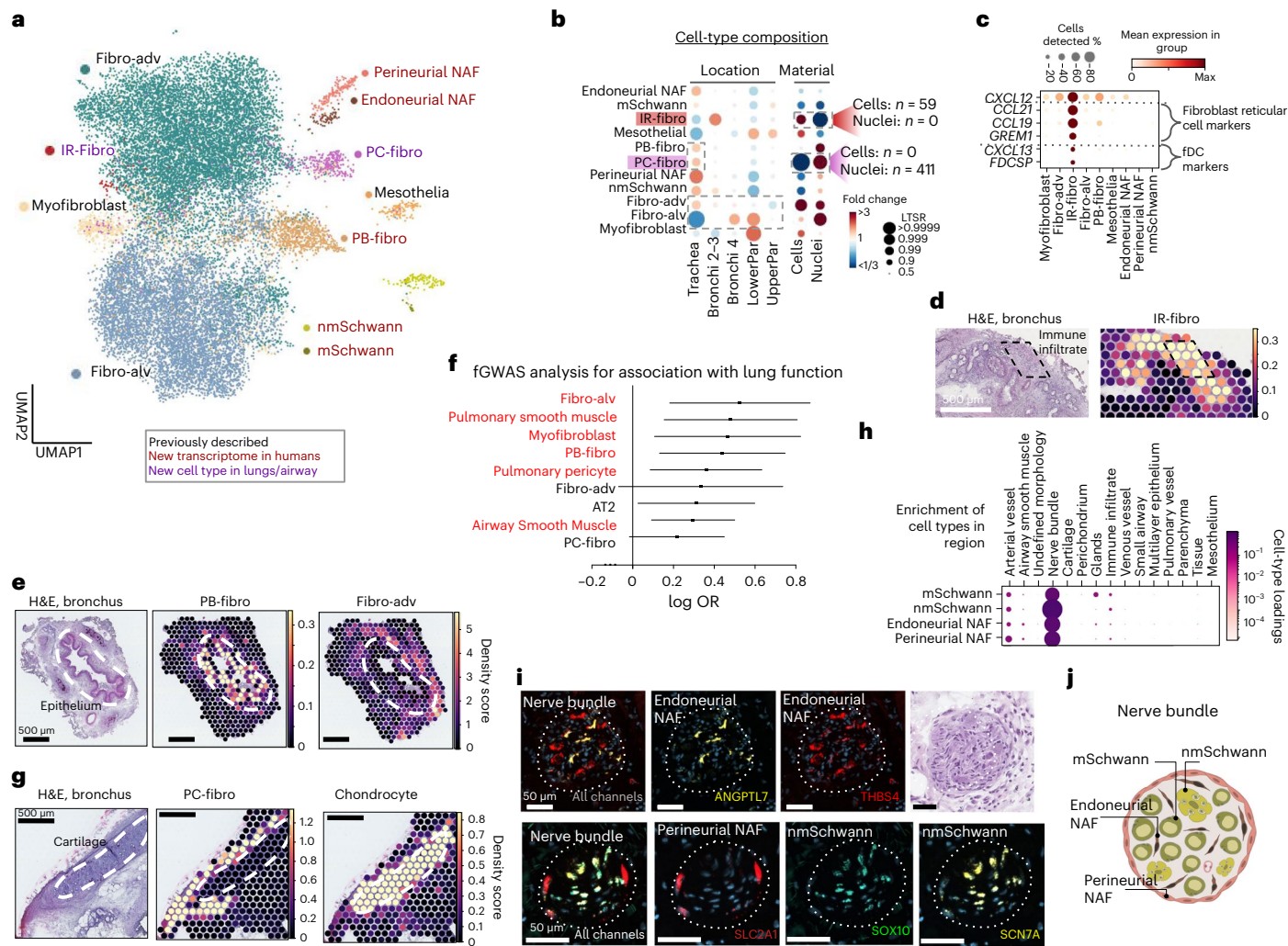

**Fig. 2 | Lung and airway fibroblasts and their spatial location. a,** Sequential clustering reveals 11 fibroblast populations in airways and lungs on UMAP, colored by novelty as shown. **b,** Sample collection location and processing method affect cell type proportions. Poisson linear mixed model analysis of cell type composition within the fibroblast compartment, accounting for location, material, dissociation protocol and donor in the model. Cell type numbers are shown in Supplementary Table 7 and in online portal. **c,** Dot plot of IR-fibro marker genes that overlap with Fibroblast reticular cell and fDC markers. **d,e,** Cell2location density scores demonstrate that (**d**) IR-fibro colocalizes with a manually annotated immune infiltrate microenvironment in the airways, and (**e**) PB-fibro localizes around the airway epithelium. **f,** PB-fibro are associated with lung function (FEV1/FVC) in fGWAS analysis (logOR, 0.53; FDR, 0.014). Shown are the log odds ratios (logOR) obtained from fGWAS and their Wald confidence

intervals (n = 19,414 genes) for several cell types. Substantially enriched cell types are marked in red (Wald test, BH multiple testing correction over 76 cell types, FDR < 0.1). **g,** Visium ST mapping of PC-fibro around the airway cartilage showing cell2location density scores. **h,** Schwann cells and NAF colocalize with peripheral nerve bundles in annotated Visium ST sections by cell2location. Cell type loadings are represented by both dot size and color. **i,** Nerve-associated cell type markers have distinct locations in the airway nerve bundles identified by smFISH staining. Donors used for replicas are shown in Supplementary Table 9. The marker gene probes for each cell type are given in each panel. Dashed lines surround the nerve bundles. **j,** Schematic representation of the described nerve-associated populations in the peripheral nerves of the airway. Fibro-alv, alveolar fibroblasts; fibro-adv, adventitial fibroblasts.

chemikines (Fig. 3e) and colocalized with a venous endothelial vessel (*ACKR1*⁺), validated in a bronchial section by Visium ST (Fig. 3f) and in smFISH microscopy (Fig. 3g and Extended Data Fig. 6f). Venous endothelial cells expressed leukocyte binding receptors (Extended Data Fig. 6g), similar to lymph node postcapillary venules, suggesting a role for venous endothelia and IR-Ven-Peri in extravasation in airway veins.

In summary, we distinguish cells of the systemic and pulmonary circulation, describe new IR-Ven-Peri cells and further define the relationship between the endothelial and perivascular cells (Fig. 3h and Supplementary Fig. 3).

**Identification of duct cells in airway SMG**

In the epithelial compartment, we identified known and rare cell types and transcriptionally define human SMG duct cells (Fig. 4a and

Extended Data Fig. 7a,b,c), enriched in the trachea[32,33] and previously only characterized in mice at the single cell level[34–36]. smFISH staining for *ALDH1A3, MIA* and *RARRES1* validated localization at the SMG and distinguished these cells from other epithelial cells (Fig. 4b and Extended Data Fig. 7e,f). Cell2location distinguished distinct locations of SMG duct cells compared to SMG mucous and serous cells, providing orthogonal evidence of the identification of a new, distinct cell type (Extended Data Fig. 7f). In addition, Velocyto analysis suggested that these cells may lie on a trajectory toward surface epithelial populations (Extended Data Fig. 7g), consistent with the regenerative role of SMG duct cells in mice[37,38].

We also identified myoepithelial cells in snRNA-seq, expressing basal epithelium (*TP63* and *KRT14*) and muscle (*ACTA2, TAGLN* and *CNN1* positive, but *DES* negative) markers (Fig. 4a and Extended Data

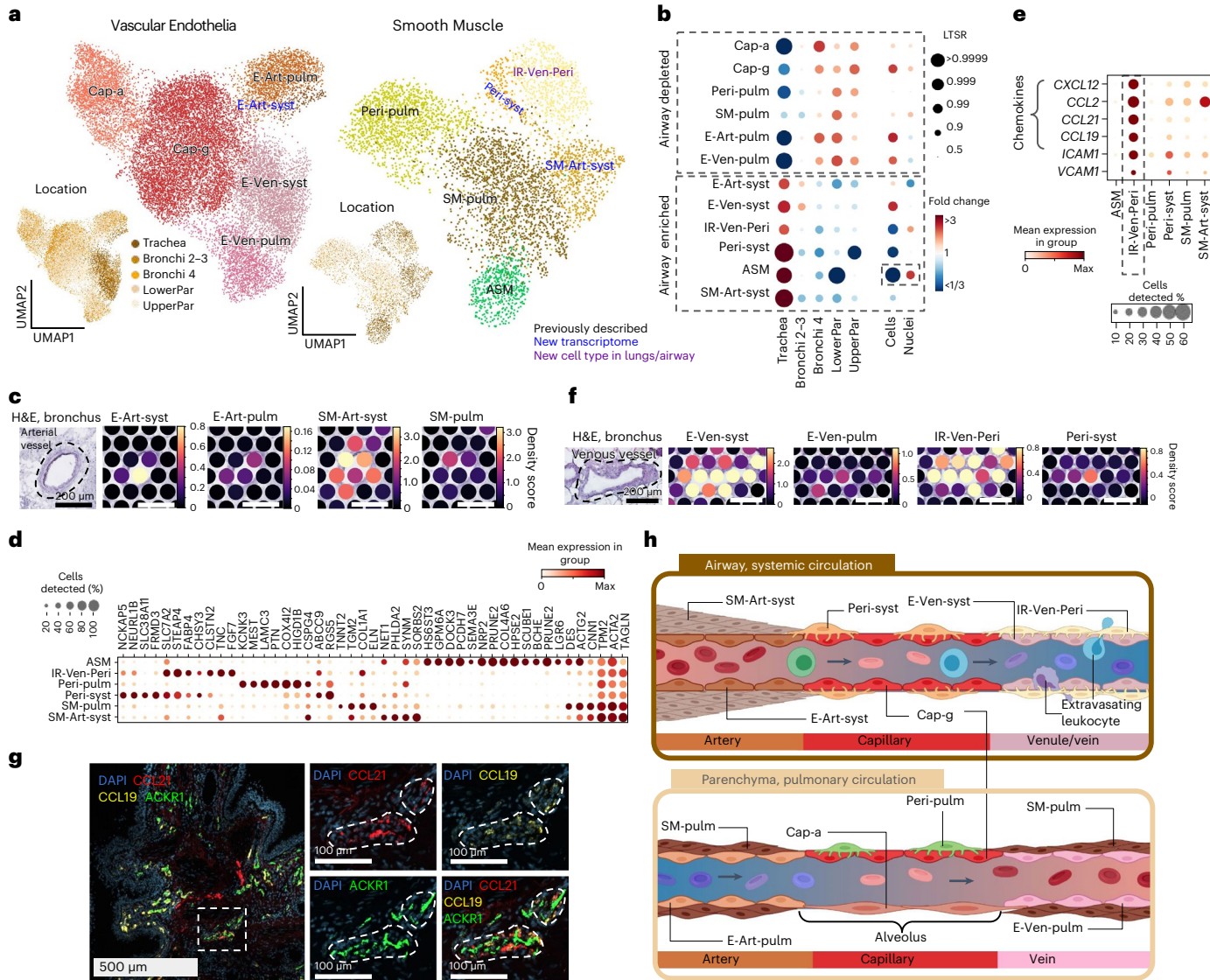

**Fig. 3 | Cell types of systemic and pulmonary circulation. a**, UMAP visualization of scRNA-seq data from the vascular endothelia and smooth muscle compartments. Color of the dots shows cell type or location. Color of the text reflects novelty. **b**, Cell type proportion analysis with fold changes and LTSR score for the cell types with regard to the location and material. Cell type numbers are shown in Supplementary Table 7 and lungcellatlas.org. **c**, E-Art-syst colocalize with arterial vessel and SM-Art-syst in the airway in Visium ST. Cell2location density scores are shown for arterial endothelial and smooth muscle cell types localizing at arterial vessel. **d**, Marker gene dot plot of the smooth muscle compartment. **e**, IR-Ven-Peri expresses immune recruiting chemokines and

cell–cell adhesion molecules shown by marker gene dot plot. **f**, E-Ven-syst and IR-Ven-Peri colocalize at a venous vessel in the airway in Visium ST sections shown by cell2location density scores at venous vessel. **g**, IR-Ven-Peri markers *CCL21* and *CCL19* localize adjacent to the venous vessel marker *ACKR1* in the airway. Donors used for replicas are shown in Supplementary Table 9. Dashed lines in **c**, **f** and **g** represent vessel structures as relevant for each figure panel. **h**, Schematic of transcriptionally defined vascular cells in the systemic and pulmonary circulation. Created with BioRender. E = endothelial, SM = smooth muscle, Cap = capillary, Ven = venous, Art = arterial, Syst = systemic, Pulm = pulmonary, Peri = pericyte, ASM = airway smooth muscle, IR = immune recruiting.

Fig. 7a,h) with localization around the glands (Extended Data Fig. 7f,i,j)[9]. We identified markers for cell–cell adhesion (*FHOD3* and *LAMA1*) and nerve synapse signaling (*NTRK2* and *PLD5*) and validated marker genes by smFISH (Extended Data Fig. 7i) and in the HPA (Extended Data Fig. 7j). Interestingly, mouse myoepithelial cells have also been shown to regenerate the surface airway epithelium[37]. However, in humans, myoepithelial cells are not well defined, potentially due to difficulties in dissociating this cell type.

Spatially, lung and airway epithelial cells were enriched in their expected manually annotated locations (Fig. 1c,d). Unbiased analysis with cell2location nonnegative matrix factorization (NMF) was able to further distinguish hidden epithelial factors. We found that basal and suprabasal cells colocate separately from apical surface epithelial cells,

consistent with their positions at the base of the surface epithelium (Extended Data Fig. 7k and Supplementary Fig. 4). AT1 cells colocalized with capillaries, alveolar macrophages and fibroblasts and were separate from AT2 cells (Extended Data Fig. 7k and Supplementary Fig. 4). This analysis revealed further spatial heterogeneity beyond manual annotations, enhancing the spatial resolution and highlighting colocating cell types.

Taking advantage of our multilocation data, we compared cells across the five locations (Extended Data Fig. 7c) and observed the expected enrichment of SMG epithelial cells and depletion of club cells in trachea (Extended Data Fig. 7c). Using our pooled snRNA-seq data, to avoid batch effects from location-specific ambient RNA contamination (Fig. 1a, pooling scheme), we also analyzed gene expression signatures.

Using a linear mixed model[39] (Methods), we detected 80 differentially expressed genes in tracheal ciliated cells, including nasopharyngeal carcinoma genes *FBXL7*, *TSHZ2* and *RAET1E* (Extended Data Fig. 7l)[40–42]. As previously reported, we found reduced *ACE2* expression in distal lung ciliated cells, where expression of *ACE2* is more relevant in AT2 cells (Extended Data Fig. 7m)[43].

Overall, we uncover the full complement of SMG epithelial cells along with their spatial contexts in the human SMG.

### Immune cells in the lung and airways
**Myeloid cells show previously undescribed heterogeneity.** We identified all major immune populations (Fig. 4c and Extended Data Fig. 8a) which were analyzed separately to reveal previously undescribed heterogeneity, especially in myeloid cells. We found known macrophage subsets, including intravascular (expressing *LYVE1* and *MAF*)[44,45], *CXC3CR*+ airway[44,46–48], *CHIT1*+[11,49] and interstitial macrophages[45]. We identified a previously undefined cluster expressing monocyte (*CD14*) and macrophage markers, termed macro-intermediate (Extended Data Fig. 8b). Among alveolar macrophages, the following two more clusters appeared: dividing cells (Macro-alv-dividing) and a cluster expressing metallothioneins (Macro-alv-MT), including *MT1G*, *MT1X* and *MT1F*. Metallothioneins have a role in binding and metabolizing metal ions[50], and in immunity and stress responses[51,52]. Finally, we identified a rare undescribed population of macrophages expressing chemokines, including *CXCL8*, *CCL4* and *CCL20*, which we named Macro-CCL[45]. The expression of *CXCL8* and *CCL20* distinguishes this subset from interstitial macrophages which express *CCL4*. *CXCL8* is associated with lung infection, asthma, IPF and COPD[53] and was identified in psoriatic skin macrophages[54].

**T and NK cell subsets in the lung and airways.** T lymphocytes and natural killer (NK) cells included CD4 T, CD8 T, mucosal-associated invariant T (MAIT), NK, NKT, innate lymphoid cells and their subsets (Fig. 4c and Extended Data Fig. 8c). In the CD4 compartment, we distinguished naive/central memory (CD4-naive/CM), effector memory/effector (CD4-EM/Effector), regulatory T cell (Treg) and tissue-resident memory (CD4-TRM) cells. Within CD8 cells, we found gamma-delta T cells (γδT), TRMs (CD8-TRM)[55] and two distinct clusters analogous to populations found in the lung in cross tissue analysis[56]: CD8-EM/EMRA and CD8-TRM/EM. The CD8-TRM cells specifically localized to airway epithelium in our spatial data (Extended Data Fig. 8d)[57,58]. NK subsets included NK-CD11d, NK-CD16hi and NK-CD56 bright[59,60]. CD11d+ NK cells are activated in response to infection in both mice and humans[61–63], were previously shown in human blood[64] and here for the first time in healthy human lungs.

T cell receptor (TCR) VDJ-seq data confirmed MAIT cell type annotation (with preferential use of *TRAJ33* and *TRAV1-2*)[65] and showed low clonal expansion in naive and Treg populations compared to memory and effector subsets (Extended Data Fig. 8e). As expected, there was no clonal sharing between individuals, but expanded clones

were found in multiple locations of the lung within a given donor (Extended Data Fig. 8f). The T and NK cell proportions displayed distinct donor-to-donor variability compared to myeloid cells (Extended Data Fig. 8g–i), consistent with higher interindividual variability in lymphocytes.

Overall, we define immune cells of the human lung and airway with unprecedented resolution.

**Colocalization of IgA plasma cells with the SMG.** B cells included naive and memory B cells, IgA and IgG plasma cells, and plasmablasts (Fig. 4c and Extended Data Fig. 9a,b). These annotations were supported by VDJ-seq B cell receptor (BCR) isotype analysis. IgA, which is important for mucosal immunity[66,67], was most frequent in the airway samples, while only the third most abundant in the parenchyma (Fig. 4d and Extended Data Fig. 9c). Distinguishing markers for IgA plasma cells included *CCR10* and B cell maturation antigen BCMA (*TNFRSF17*; Extended Data Fig. 9b), which are important for plasma cell localization and survival, respectively[67–69].

In Visium ST data, IgA plasma cells mapped to the airway SMG, colocalizing with duct, mucous and serous cells, while IgG mapped to immune infiltrates (Fig. 4e). Enrichment of plasma cells (MZB1+) at the SMG was confirmed in the HPA (Extended Data Fig. 9d,e), building on a study in the 1970s that first showed IgA plasma cells in human airway SMG[70]. We further distinguished enrichment of IgA plasma cells in the serous glands with cell2location NMF, showing two distinct gland factors, one with SMG serous cells colocalizing more with IgA plasma cells than a second distinct factor with other SMG epithelial cells (Fig. 4f, Extended Data Fig. 9f and Supplementary Fig. 4). This preferential localization of IgA plasma cells was confirmed by manual annotation of gland areas in ST on formalin-fixed paraffin-embedded (FFPE) preserved tissue samples, which allowed better distinction of serous and mucous glands (Extended Data Fig. 9g).

To dissect this niche at single-cell resolution, we used multiplex IHC to confirm the presence of IgA2 but the absence of IgG cells in the SMG (Fig. 4g and Extended Data Fig. 9h), consistent with Visium ST (Fig. 4e,f). We also detected IgD+ naive B cells and CD3+ CD4+ T helper cells in the human SMG (Fig. 4g). We hypothesize that together these different cell types constitute an immune niche with relevance in disease, which we term the gland-associated immune niche (GAIN). Mucosal IgA is important for protection against respiratory infections[2], and we found that proportions of IgA plasma cells were increased in coronavirus disease 2019 (COVID-19) patients versus healthy controls in single-cell data from published nasal, tracheal and bronchial brush samples (Methods) (Fig. 4h)[71]. In addition, increased plasma cell numbers have been shown in smokers[70], patients with cystic fibrosis[72], COPD[73] and Kawasaki disease[74], warranting further study of the GAIN in these conditions. In C57/BL6 mouse tracheal sections, we did not identify IgA+ cells in the SMG of two independent cohorts of mice despite IgA+ staining in the colon as expected[75] (Extended Data Fig. 9i), suggesting that the GAIN should be studied in humans.

---

**Fig. 4 | IgA plasma cells in human airways colocalize with SMGs. a**, UMAP of scRNA-seq and snRNA-seq data from epithelial cells (excluding alveolar AT1 and AT2) colored by cell type, with a dot plot for the marker genes of SMG duct cells. **b**, smFISH staining of mucous (*MUC5B*), serous (*LPO*) and duct (*ALDH1A3/RARRES1*) cells in human bronchus. **c**, UMAP plots of myeloid, T/NK and B lineage cells, colored by cell type. **d**, Number of B lineage cells with different Ig isotypes in airway (trachea and bronchi) from the analysis of VDJ amplified libraries. **e**, Visium ST results show IgA plasma cells specifically localize in the glands. Normalized average cell abundance (dot size and color) are shown from cell2location for SMG and B lineage cell types across the manually annotated micro-anatomical tissue environments. H&E section of bronchi with manually annotated glands shown in blue (top panel), cell2location density scores for IgA plasma cells and SMG serous cells shown in lower panels. Cell type loadings are represented by both dot size and color. **f**, Unsupervised NMF analysis of Visium ST cell2location results for 11 factors, showing NMF factor loadings normalized per cell type (dot size and color). Factors 3 and 6 identify two separated factors in the SMG colocalizing IgA plasma cells, specifically with SMG serous cells (factor 3), but less with other SMG cell types (factor 6). Other factors and cell types are shown in Supplementary Fig. 4. **g**, Multiplex IHC staining of human trachea for the SMG structure (Hoechst for nuclei, EpCAM for epithelium, Phalloidin for actin, CD31 for vessels), B lineage markers (IgD, IgA2 and IgG) and CD4 T cells (CD45, CD3 and CD4). Arrowheads point to CD45+ CD3+ CD4+ cells. Scale bar 100 μm. **h**, Percentages of different isotypes of 470 B plasma cells from nasal, tracheal and bronchial brushes of two COVID-19 positives and three healthy control patients[71]. Patients with over 20 plasma cells were considered. Donors used for replicas in **b** and **g** are shown in Supplementary Table 9. Macro, macrophage; CM, central memory; EM, effector memory; EMRA, effector memory re-expressing CD45RA.

**Cell–cell interactions and the SMG immune cell niche.** To understand colocalization of B cells, IgA plasma cells and T cells in the SMG (Fig. 4e–g), we explored the molecular mechanisms underpinning the GAIN. Expression of *PIGR*, which transcytoses polymeric Ig across the surface epithelium, was high across SMG epithelial cells, as was *CCL28*, known to recruit IgA plasma cells through *CCR10* (Fig. 5a–d and

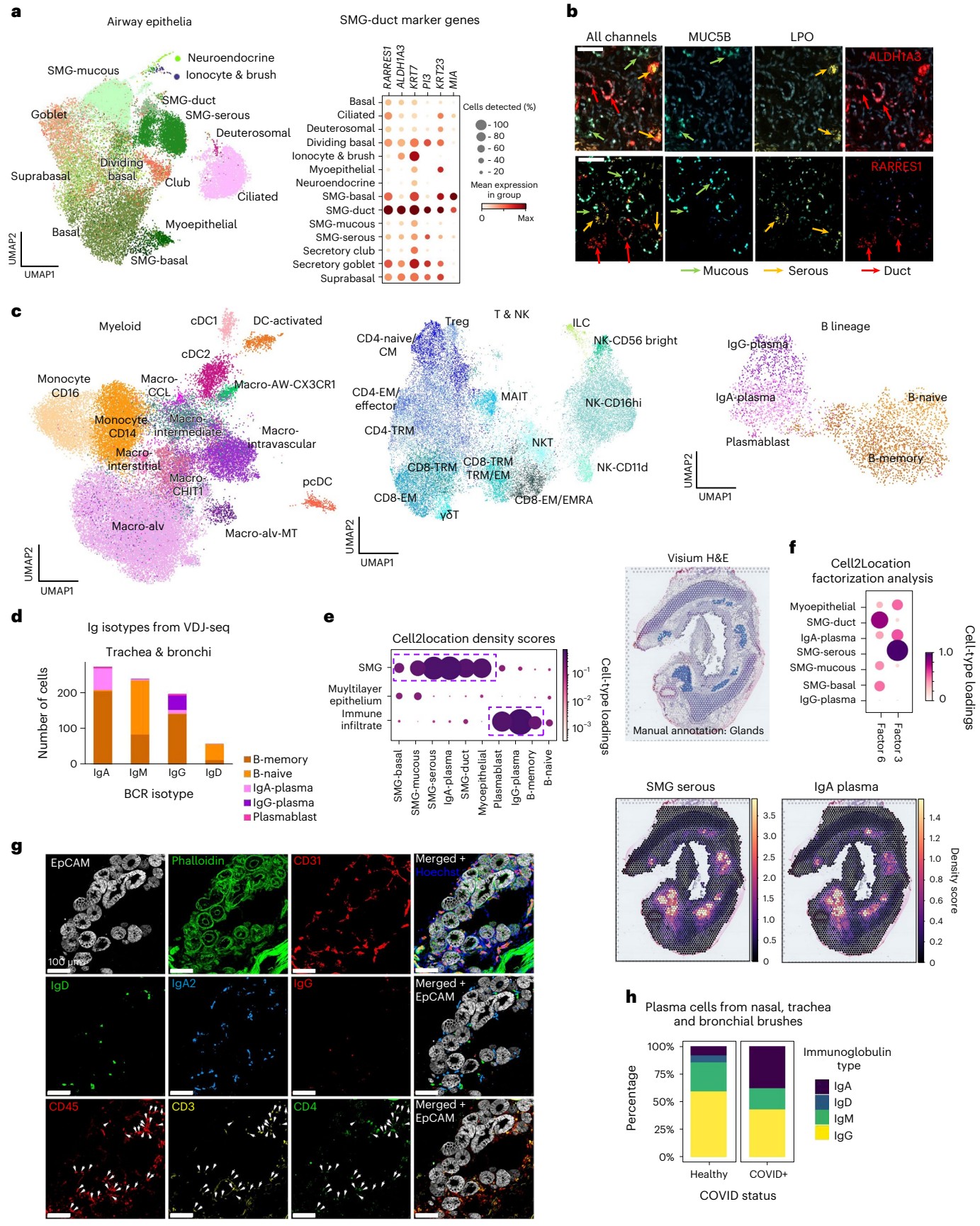

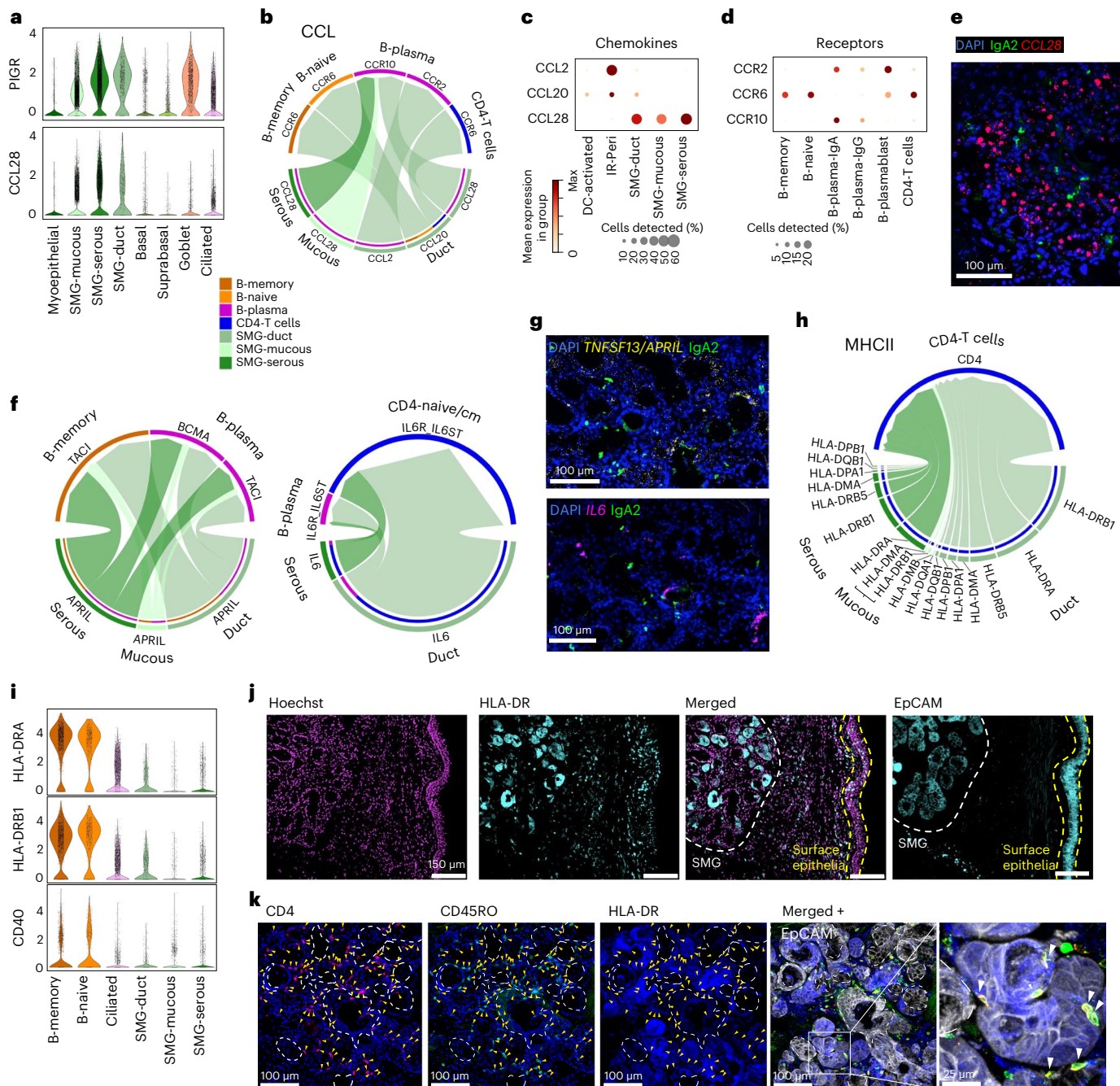

**Fig. 5 | Cell–cell signaling at the SMG for B cell recruitment and survival.**
**a**, Expression of *PIGR* and *CCL28* in epithelial cells by violin plot. **b**, CellChat cell–cell interaction analysis pathways for CCL chemokines produced by SMG epithelial cells and received by B cells (memory, naive and IgA/IgG/plasmablast combined) or CD4 T cells (CD4-naive/CM, CD4-EM/Effector and CD4-TRM combined) within airway tissue (trachea and bronchi). Arrow direction denotes chemokine-receptor pairs on specific cell types, arrowhead thickness reflects the relative expression of chemokine signal from each cell type. **c,d**, Expression dot plot of relevant chemokines and corresponding receptors as shown in **b. e**, smFISH (*CCL28*) and IHC (IgA2) staining in tracheal SMG. **f**, CellChat analysis as in **b** showing signaling of *TNFSF13*/APRIL and *IL-6* from SMG epithelial cells to relevant B cell subsets and CD4-naive/CM. The proportion of the circle for each gene/cell type reflects the relative expression. **g**, smFISH (*IL-6*, *TNFSF13*/APRIL)

and IHC (IgA2) staining in tracheal SMG. **h**, CellChat analysis as in **f** showing signaling from *HLA* genes expressed by SMG epithelial cells, signaling to CD4 on CD4 T cells. **i**, RNA expression of *HLA-DRA*, *HLA-DRB1* and *CD40* in B cells (as professional antigen-presenting cells for comparison) on violin plot, ciliated and SMG epithelial cells from scRNA-seq/snRNA-seq. **j**, IHC of HLA-DR and EpCAM in human airways showing strong expression of HLA-DR in the SMG (white dashed line) compared to the surface epithelium (yellow dashed line). **k**, IHC staining of CD4, CD45RO, HLA-DR and EpCAM, as indicated, in the airway SMG showing localization and close contact of CD4+ CD45RO+ T cells with HLA-DR+ glands as shown in the enlargement. Dotted lines enclose HLA-DR negative or low regions of glands, yellow arrowheads denote CD4+ CD45RO+ T cells, white arrowheads in the zoom-in show CD4+ CD45RO+ cells interacting with HLA-DR+ gland epithelial cells. Donors used for replicas in **e**, **g**, **j** and **k** are shown in Supplementary Table 9.

Extended Data Fig. 10a)[68,76]. We confirmed expression of *CCL28* in SMG by smFISH, and at the protein level by IHC (Fig. 5e and Extended Data Fig. 10b), and observed a gradient of expression along the proximal

to distal axis, where *CCL28* is highest in SMG duct and serous cells of the trachea (Extended Data Fig. 10b,c). This gradient in serous cells was statistically significant ($P < 0.05$) with Spearman's two-tailed rank

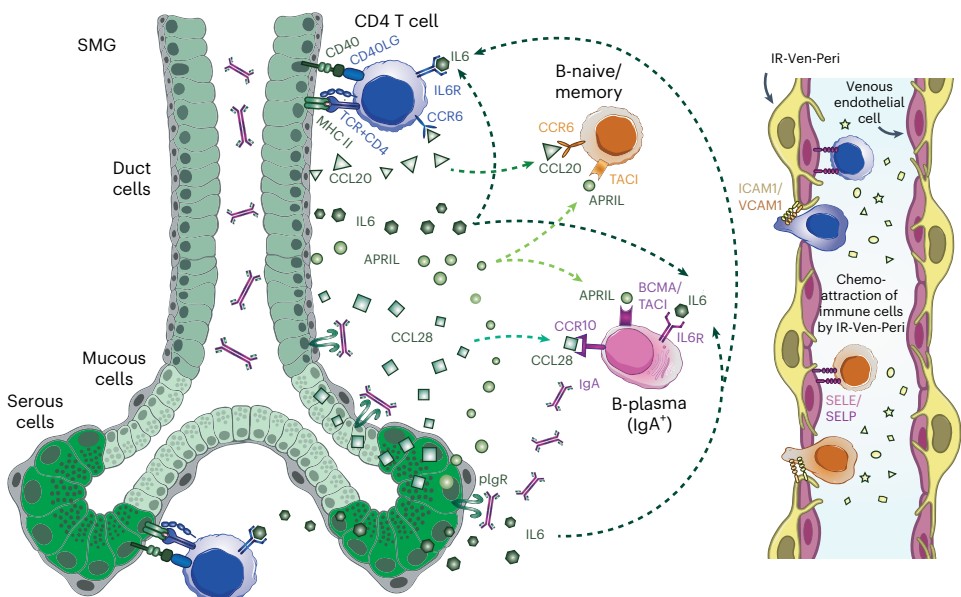

**Fig. 6 | Schematic of the human airway GAIN.** Schematic of the GAIN showing immune cell recruitment and extravasation facilitated by venous endothelial cells and IR-Ven-Peri (immune recruiting venous perivascular cells) and signaling patterns between SMG epithelial cells, CD4 T cells, B naive/B memory cells and B plasma cells to attract immune cells and promote antigen-specific T cell-dependent and T cell-independent pathways, leading to IgA secretion at the SMG.

correlation analysis. Cell–cell interaction analysis using CellChat[77] on cells from the airways again confirmed the *CCL28-CCR10* axis between SMG epithelial cells and B plasma cells (combined IgA, IgG and plasmablasts) and predicted SMG duct cells to recruit memory and naive B cells and CD4 T cells (combined CD4 subsets, excluding Tregs) through *CCL20* (refs. [78–80]) (Fig. 5b–d).

In addition to immune cell recruitment, we explored signals supporting B and plasma cell function in the GAIN. A proliferation-inducing ligand (*APRIL*), a factor important for B cell survival, differentiation and class switching, was expressed by SMG duct/serous cells interacting with the receptors *TACI* and *BCMA* on B cells (Fig. 5f and Extended Data Fig. 10a). smFISH for *APRIL* in tissue confirmed expression in glands, especially in serous cells (*LPO*[+]*RARRES1*[−] *APRIL*[high]), confirming the specific B and IgA colocalization from our ST analysis in the *APRIL*[high] serous glands (Fig. 5g and Extended Data Fig. 10d,e). Interestingly, *APRIL* expression can be induced on intestinal epithelial cells leading to IgA2 class-switch recombination (CSR) in the local tissue environment[81]. We found activation-induced cytidine deaminase (*AICDA*) expression in a few B memory cells, suggesting the possibility of local CSR at the SMG (Extended Data Fig. 10f).

In combination with *APRIL*, *IL-6* induces and supports long-lived plasma cells, potently induces IgA secretion[82,83] and increases IgA secretion in COPD[84]. In our data, SMG duct/SMG serous cells expressed *IL-6* (Fig. 5g), which was predicted to interact with *IL-6R/IL-6ST* on B plasma and CD4-naive/CD4-CM T cells (Fig. 5f and Extended Data Fig. 10a). IL-6 has been shown as a required factor for CD4 T cell memory formation and for overcoming Treg mediated suppression[85]. Salivary gland epithelial cells are known to induce B cell responses via *IL-6* both directly (T cell-independent) and via T cell-dependent mechanisms[86]. *IL-6* is also upregulated in serum and bronchoalveolar lavage fluid in asthma and COPD patients, suggesting the importance of GAIN in disease[87–89].

CellChat also predicted interactions between *HLA* genes expressed by SMG epithelial cells, and CD4 T cells (Fig. 5h,i) that indicate antigen presentation directly by the SMG epithelial cells. The expression of *HLA-DRA* and *HLA-DRB1* in SMG duct/SMG serous cells was comparable to ciliated cells at the RNA level (Fig. 5i), but much higher at the protein level (Fig. 5j). Similarly, the costimulatory gene *CD40* was expressed in SMG epithelial cells (Fig. 5i). CD4 T cells also localized to HLA-DR[high] nonmucous regions of glands (Fig. 5k and Extended Data Fig. 10g,h). CD4 T cells in the glands were CD45RO[+] (memory) cells, and could be seen closely interacting with HLA-DR[high] SMG epithelial cells, suggesting direct cell–cell contact (Fig. 5k and Extended Data Fig. 10i–k). Overall, our data suggest that SMG Serous/duct cells can present antigen to CD4 T cells, similar to airway and nasal epithelial cells, which can promote T cell proliferation in vitro[90–94]. Antigen presentation by SGP[low]MHCII[high] epithelial cells in the parenchyma of mice has been shown to regulate CD4-TRM responses, contributing to immune homeostasis[95]. MHCII[high] SMG epithelial cells may have a similar function in the airways.

In conclusion, we identified the colocalization of IgA plasma cells, naive/memory B cells and T cells at the serous glands and described molecular signaling pathways for the recruitment and maintenance of immune cells at the SMG. The described pathways are functional in secondary lymphoid structures such as MALT, and we now suggest they can establish the GAIN (Fig. 6) of the human airways.

## Discussion

By integrating scRNA-seq/snRNA-seq with ST, we provide fine-grained resolution of 80 cell types/states in the human lung and airways. Eleven previously unannotated cell types/states were identified and mapped to distinct micro-anatomical tissue environments. Our data have contributed to the HLCA[10], which provides means for assessing intraindividual variation and effects of clinical covariates. Our in-depth study reveals airway tissue niches encompassing previously unresolved cell types and their interactions, as well as unexpected properties of cell signaling relationships. We provide transcriptomic profiles of human airway

chondrocytes, cells of the peripheral nerve bundles, SMG duct cells, enhanced resolution in fibroblast, macrophage and lymphocyte subsets and distinguish between pulmonary versus systemic vasculature and pericytes. We highlight potential disease associations for some of these new populations, such as PB-fibro for COPD and IPF. Finally, we discover the GAIN with likely relevance in inflammatory and infectious diseases. We present these data as a resource to the community as open-access downloadable files and through our interactive web portal (lungcellatlas.org).

The GAIN defines an immunomodulatory role for SMG epithelial cells, which are central to signaling circuits for local IgA responses. Specifically, we highlight that IgA plasma cells, B cells and CD4 T cells are recruited via chemokines secreted by IR-pericytes and SMG duct/serous cells. The survival, maturation and, potentially, class switching of B lineage cells are supported by *APRIL* and *IL-6*, providing T cell independent factors. Additionally, SMG duct/serous cells have the potential to induce and/or modulate antigen-specific responses through the expression of MHC-II and *CD40*. Many of these pathways have been observed in other tissues, particularly in the salivary glands and within secondary lymphoid tissues such as Peyer's patches. No such secondary lymphoid structures have been observed within healthy airways. We hypothesize that GAIN is an important site for local induction of immune responses and homeostasis.

This newly defined IgA immune niche is likely to play an important role in common lung diseases as well as respiratory infections—IgA plasma cells are increased in the airways of COPD[84] and patients with cystic fibrosis[72], and pIgR-mediated IgA transport is dysregulated in asthma[96] and pulmonary fibrosis[97]. In patients with COVID-19, early severe acute respiratory syndrome coronavirus 2 (SARS-CoV-2) neutralization was more closely correlated with IgA than IgM or IgG[98] and we report higher proportions of IgA plasma cells in the airways of patients with COVID-19. Nasal vaccines can induce a strong local sIgA response and prevent viral shedding[99] in the respiratory tract[100]. A better understanding of the GAIN is therefore highly relevant for maintaining lung health and providing immunity to respiratory infections such as COVID-19.

## Online content

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

[1]Wellcome Sanger Institute, Wellcome Genome Campus, Cambridge, UK. [2]European Molecular Biology Laboratory, European Bioinformatics Institute (EMBL-EBI), Wellcome Trust Genome Campus, Cambridge, UK. [3]Molecular Immunity Unit, University of Cambridge Department of Medicine, MRC Laboratory of Molecular Biology, Francis Crick Ave, Cambridge, UK. [4]Department of Genetics and Evolutionary Biology, Institute of Biosciences, University of São Paulo, São Paulo, Brazil. [5]Kindai University Faculty of Pharmacy, Higashi-osaka, Japan. [6]UCL Respiratory, Division of Medicine, University College London Hospitals NHS Foundation Trust, London, UK. [7]Department of Surgery, University of Cambridge, and Cambridge NIHR Biomedical Research Centre, Cambridge, UK. [8]European Molecular Biology Laboratory (EMBL), Heidelberg, Germany. [9]Deutsches Krebsforschungszentrum (DKFZ), Heidelberg, Germany. [10]Theory of Condensed Matter, Cavendish Laboratory/Department of Physics, University of Cambridge, Cambridge, UK. [11]These authors contributed equally: Elo Madissoon, Amanda J. Oliver. ✉e-mail: st9@sanger.ac.uk; km16@sanger.ac.uk

 

## Methods

### Experimental methods

**Access to human tissue and ethics oversight.** Samples were obtained from deceased transplant organ donors by the Collaborative Biorepository for Translational Medicine (CBTM) with informed consent from the donor families and approval from the National Research Ethics Services (NRES) Committee of East of England, Cambridge South (15/EE/0152). CBTM operates in accordance with UK Human Tissue Authority guidelines.

**Tissue dissociation and single-cell sequencing.** Tissue was collected from 13 donors from five lung locations including trachea, bronchi at the second/third generation, bronchi at the fourth generation, upper left lobe parenchyma and lower left lobe parenchyma (Fig. 1a and Supplementary Tables 1–3). In total, 11 donors were profiled for fresh and frozen transcriptomic analysis with two additional donors used for FFPE ST and smFISH/IHC validation (summarized in Supplementary Table 9). Following collection at the clinic, samples (range: 1–4 cm$^3$) were immediately placed into cold Hypothermasol FRS[32]. Within 12 h after circulation ceased, samples were dissociated (seven donors; Supplementary Tables 1 and 9) and/or preserved in optimal cutting temperature (OCT) compound and frozen in isopentane at −60 °C for later spatial analysis (six donors; Supplementary Table 3) and nuclei isolation (seven donors; Supplementary Tables 2 and 9). Most samples (*n* = 5) were digested using liberase and trypsin, and CD45 positive cells loaded on 10X as a separate fraction (protocols.io.39ygr7w). One donor was digested with collagenase D for comparison (protocols. io.34kgquw; Supplementary Tables 1 and 2). Briefly, tissue dissociation used (for five donors) 1 g of lung tissue washed with PBS, minced finely with scalpels, before treatment with 13 U ml$^{-1}$ liberase TL and 0.1 mg ml$^{-1}$ DNase I for 30 min at 37 °C with rocking. Cells were filtered through a 70 μm strainer, washed with neutralization media (RPMI$^+$ 20% FBS) and pelleted (sample P1). Tissue remaining in the cell strainer was digested with 0.25% trypsin-EDTA with DNase I for 30 min at 37 °C with rocking, filtered and washed with neutralization media. Meanwhile, sample P1 was treated with red blood cell lysis buffer before being separated into CD45 positive and negative fractions using MACS (Miltenyi, as according to the manufacturer's protocol). The CD45 negative fraction was pooled with cells from trypsin treatment, resulting in the following two samples for loading on 10X: CD45 positive cells from liberase TL digestion (to enrich for immune cells) and pooled CD45 negative liberase-treated cells with trypsin-treated cells (nonimmune fraction). Both fractions were resuspended in 0.04% BSA/PBS, counted and loaded on the 10X Genomics Chromium Controller, aiming to capture 5,000 cells, according to the manufacturer's protocol. The 10X Genomics chemistry is included in Supplementary Table 1.

**Single-nucleus sequencing.** Our single nuclei isolation method[101] from frozen tissue (Supplementary Table 2) used 8 × 50 μm thick sections which were homogenized using a glass Dounce homogenizer (Sigma) in nuclei isolation buffer (NIM; 0.25 M sucrose, 0.005 M MgCl$_2$, 0.025 M KCl, 0.01 M Tris (buffer pH7.4), 0.001 M DTT and 0.1% Triton X-100) in the presence of Complete protease inhibitors (Roche) and RNAse inhibitors RNasin (Promega)—0.4 U μl$^{-1}$ and SUPERase-In (Invitrogen) 0.2 U μl$^{-1}$). Tissue was homogenized using ~15 strokes with pestle A (clearance 0.0028–0.0047 in.) and then pestle B (clearance 0.0008–0.0022 in.). Isolated nuclei were filtered through a 40 μM filter, collected at 2,000*g* and resuspended in 0.5 ml of storage buffer (PBS containing 4% BSA and RNasin (Promega)—0.2 U μl$^{-1}$). Nuclei were incubated with NucBlue (ThermoFisher) and purified from debris by FACs sorting, stained with Trypan blue and counted. Five thousand nuclei from five different samples were pooled and all 25,000 nuclei were loaded onto the 10X chromium controller using the 3′ v3.1 kit as per the Chromium Single Cell 3′ Reagent Kits v3 User Guide, targeting to recover ~3,000 nuclei per sample. Post-GEM-RT cleanup, cDNA

amplification and 3′ gene expression library construction were performed according to the user guide and libraries were sequenced on the Novaseq platform.

**Spatial transcriptomics.** Samples ≤0.5 cm$^2$ were cut from the five lung and airway locations outlined above. Most of the parenchyma tissue was removed from bronchi samples, which were embedded in OCT and flash frozen in −60 °C isopentane (for six donors; Supplementary Table 9) or fixed for 24 h in 10% neutral buffered formalin and processed into wax (FFPE, for one donor; Supplementary Table 9). H&E staining was used to determine the morphology of tissue blocks before proceeding with ST. Sections of 10 μm (fresh frozen samples) or 5 μm (FFPE) were then cut from the blocks onto Visium slides (10X Genomics) and processed according to the manufacturer's protocol. Further details on samples are in Supplementary Table 3 and Supplementary Fig. 1. H&E images generated during the Visium protocol were captured at ×20 magnification on a Hamamatsu Nanozoomer S60.

Dual-indexed libraries were prepared as in the 10X Genomics protocol, pooled at 2.25 nM and sequenced (four samples/Illumina Novaseq SP flow cell) with read lengths 28 bp R1, 10 bp i7 index, 10 bp i5 index, 90 bp R2 for fresh frozen samples or 50 bp R2 for FFPE.

**smFISH.** smFISH was performed in multiple sections from at least two donors (Supplementary Table 9). Tissue blocks for smFISH (RNAScope) in situ hybridization were chosen based on H&E staining. Ten micron-thick cryosections cut onto superfrost plus slides were processed using the RNAScope 2.5 LS multiplex fluorescent assay (ACD, Bio-Techne) on the Leica BOND RX system (Leica). Fresh frozen lung sections were fixed for 90 min with chilled 4% paraformaldehyde, washed twice with PBS and dehydrated through an ethanol series (50%, 70% and 100% ethanol) before processing according to the manufacturer's protocol with protease IV treatment. Samples were first tested with RNAScope positive and negative control probes before proceeding to run probes of interest. Slides were stained for DAPI (nuclei) and three to four probes of interest, with fluorophores Opal 520, Opal 570, Opal 650 and ATTO 425 at between 1:500 and 1:1,000 concentration. These were then imaged on a Perkin Elmer Opera Phenix high-content screening system with water immersion at ×20 magnification. Imaging data were processed using Omero (Open Microscopy Environment).

**Mouse samples.** Wild-type C57/BL6 mouse samples were obtained from Kindai University, Japan (courtesy of T. Nakayama) and Charles River, USA (AMSbio). Male (colon samples) and female (all other samples) mice were maintained in specific pathogen-free conditions and used at 8- to 10-week old. All animal experiments for mice obtained from Kindai University were approved by the Centre of Animal Experiments at Kindai University. Mouse tissue from Charles River was purchased from a certified animal supplier through AMSbio, with an internal ethical approval process for broadly defined research use.

**Tissue preservation and antibody staining.** For IBEX staining, the fresh airway tissue was received in cold Hypothermasol, fixed with 1% PFA solution for 24 h at 4 ˚C and transferred to cold 10% and 30% sucrose gradient for ~8 and ~12 h, respectively, before freezing in OCT. The fixed tissue was sectioned at 10–30 μm thickness. Iterative staining of human trachea sections was performed as described by Radtke et al. (ref. [102]). Sections were permeabilized and blocked in 0.1 M Tris, containing 0.1% Triton (Sigma), 1% normal mouse serum, 1% normal goat serum and 1% BSA (R&D). Primary antibodies were incubated for 2 h at room temperature and secondary for 1 h at room temperature in a wet chamber, washed three times in PBS and mounted in Fluoromount-G (Southern Biotech). Images were acquired using a TCS SP8 (Leica) inverted confocal microscope. The coverslip was removed, slides were washed three times in PBS and fluorochromes were then bleached using a 1 mg ml$^{-1}$ solution of lithium borohydride in water (Acros Organics) for

15 min at room temperature. Slides were washed in PBS (three times) before repeating staining, up to a total of five rounds of staining. Raw imaging data were processed using Imaris (Bitplane) using Hoechst as fiducial for the alignment of subsequent images. The staining setup and antibody information are in Supplementary Table 6.

For human airway samples and mouse trachea/colon samples, costained for CCL28 and IgA2/IgA (Supplementary Table 6), 10 μm thick OCT embedded fresh frozen sections were fixed with cold acetone (human) or room temperature acetone:ethanol (1:1) (mouse) for 20 min, followed by blocking in the buffer above (human) or 2% BSA/PBS with 1:800 rat serum (Abcam) (mouse) for 1 h at room temperature. Primary antibodies were incubated in blocking buffer for 1 h at room temperature and washed three times in PBS. Secondary antibodies were incubated for 1 h at room temperature followed by another three times PBS washes and 5 μg ml$^{-1}$ DAPI (Invitrogen) for 5 min before coverslipping with ProLong Gold Antifade Mountant (LifeTechnologies). Slides were imaged using Hamamatsu S60 slide scanner at ×40 magnification. Imaging data were processed using Omero (Open Microscopy Environment).

## Computational analysis

**Mapping of gene expression libraries.** scRNA-seq and snRNA-seq gene expression libraries were mapped with Cell Ranger 3.0.2, and Visium libraries were mapped with Space Ranger 1.1.0 from 10X Genomics (https://support.10xgenomics.com). Both types of libraries were mapped to an Ensembl 93-based reference (10X-provided GRCh38 reference, version 3.0.0). For nuclei samples, the reference was altered into a pre-mRNA reference as per 10X instructions. TCR/BCR libraries were mapped with Cell Ranger 4.0.0 to the 10X-provided VDJ reference, version 4.0.0.

**scRNA-seq and snRNA-seq analysis.** The CellRanger unfiltered matrices were used as an input for the SoupX v1.0.0 algorithm[103] to remove ambient RNA contamination, according to the tutorial (https://github.com/constantAmateur/SoupX). For each snRNA-seq library, CellRanger filtered nuclei subjected to QC filters, pass QC nuclei were processed using standard scanpy pipeline and were clustered to form five to ten clusters. Nuclei that did not pass QC were assigned to those clusters by logistic regression. This clustering was then passed to SoupX, to derive a set of cluster-specific genes for automatic estimation of contamination rate. Default values were used for SoupX's functions, except for 'autoEstCont()' where 'soupQuantile' was set to 0.8. The single-cell and nuclei libraries with SoupX correction were analyzed using the standard scanpy 1.7.1 workflow[104]. The cells with >4,000 counts in nuclei and 20,000 counts in the cells were removed. In the cells, droplets with >10,000 features were removed. Lower threshold of 1,000 features was applied to donor A37 due to difficulty to remove ambient RNA contamination. Master cell types were annotated and extracted for reanalysis with scanpy workflow, including new highly variable genes (HVGs) detection. Between 1,000 and 3,000 HVGs were used to define 40 principal components for calculating the UMAP. Data integration via Harmony v1.0 (ref. [105]), BBKNN v1.4.1 (ref. [106]) or scVI-tools v0.9.0 (ref. [107]) was used with either 'material' (cells versus nuclei), or 'material and donor'. Doublet clusters, identified by observing markers from multiple cell types and higher counts in snRNA-seq, were removed. An iterative clustering approach was used to derive clusters for less abundant cell types. In addition to known marker genes, new ones were derived using the scanpy rank_genes_groups function. Cell types and master clusters were annotated according to known and newly derived markers as in Supplementary Note 1 and in consensus with other studies in Supplementary Table 4. The UMAP of the airway epithelial cell types was achieved by integrating the data with published human airway epithelial cells[9], including previously unannotated serous and mucous cells identified from the unprocessed data of their study. Altogether we detected all cell types from at least two different locations and five different donors. The distribution of cell types with respect to location, material and donor variables are shown in Supplementary Table 7 and visualized by the contribution of donor in Supplementary Fig. 2.

**Spatial mapping of cell types using Visium ST and cell2location.** Visium ST data was analyzed by integrating scRNA-seq/snRNA-seq and spatial transcriptomes with the cell2location method (v0.1)[13]. Cell2location estimates reference gene expression signatures of cell types from scRNA-seq using Negative Binomial regression that accounts for batch effects. In Extended Data Fig. 3b, we extended the cell type reference to germinal center cell types from a published human gut dataset[18]. Cell2location uses the reference signatures to estimate absolute spatial abundance of cell types, integrating and normalizing data across 11 fresh frozen Visium sections (five were excluded based on quality control metrics). 10X Visium data were processed to untransformed and unnormalized mRNA counts, filtered to genes shared with scRNA-seq, with hyperparameters in cell2location based on the tissue and experiment quality as follows:

(1) Expected cell abundance per location $\hat{N} = 20$
(2) Regularization of within-experiment variation in RNA detection sensitivity of $\alpha^y = 20$

The model was trained until convergence (40,000 iterations). Loss function (ELBO) scaling by locations × genes was used. Pearson correlation between $\log_{10}(x+1)$-transformed observed mRNA counts and expected mRNA amount from the cell2location model assessed the model quality (Pearson $R = 0.745$).

Micro-anatomical tissue environments were labeled from the H&E images. Only Visium spots aligned and annotated as 'tissue' were used for analysis (manual annotation). Specific tissue environments are listed in Supplementary Table 5 and at lungcellatlas.org loupe browser. Visium FFPE (four sections) allowed better conservation of morphology and therefore had more detailed manual annotations including the separation of mucous glands from the seromucous/other glands. The annotation of mucous-only glands was conservative to distinctly separate mucous-only glands, as the transcripts from high-count serous cells contaminated neighboring spots. The manual annotations were used to compute cell abundance of each cell type across micro-environments.

We further used NMF of cell abundance estimated by cell2location for unbiased microenvironment identification. Scikit-learn NMF implementation from cell2location package was used. NMF was trained with a range of factor numbers (8–24). NMF factor loadings for cell types are reported in the paper as dot plot normalized per cell type by the sum of NMF loadings, which can be interpreted as a proportion of cells of each cell type present in each tissue zone. NMF factor loadings across locations are reported as the total cell abundance of constituent cell types.

**Postprocessing analysis.** Gene set enrichment analysis was performed in GSEA online tool (https://www.gsea-msigdb.org/gsea/index.jsp)[108] for specific gene sets and in gProfileR e106_eg53_p16_65fcd97 (https://biit.cs.ut.ee/gprofiler/gost). The analysis of gene expression in GTEx tissues was performed in the GTEx portal (https://gtexportal.org/home/). Cell–cell interaction analysis was performed with CellChat (http://www.cellchat.org/)[77]. To reduce donor-to-donor differences, the dataset was downsampled to a set number of cells per donor per cell type.

The code used for marker gene dot plots with mean group expressions and expression of TCR regions was previously published with code available at https://doi.org/10.5281/zenodo.3711134.

Pseudotime analysis for selected cell populations was performed with Monocle3 (ref. [109]), its functionality to infer a pseudotime based on UMAP coordinates. The root was identified as the cell with the highest combined expression of canonical progenitor markers (*VCAN* for chondrocytes; *TGM2*, *HMCN2* and *SULF1* for smooth muscle).

Cell trajectory analysis was performed using the scVelo package (v0.2.1)[110] and specifying the stochastic model.

Label transfer was performed via Azimuth tool v0.4.1 (https://azi-muth.hubmapconsortium.org/) with the lung reference data[11] v1.0.0. CellTypist[56] v1.2.0 was used to train a logistic regression model from our dataset for label transfer using the *celltypist.train* function (tag-value pairs: use_SGD=True, feature_selection=True). HPA (https://www.pro-teinatlas.org/humanproteome/tissue) was used for extracting images of protein stainings with antibodies on human tissues.

**BCR and TCR analysis from VDJ-data.** VDJ analysis was done with Scirpy 0.6.0 (https://icbi-lab.github.io/scirpy/)[111]. For TCR data, clonotypes were defined based on CDR3 nucleotide sequence identity. For BCR data, clonotypes were defined based on the Hamming distance between CDR3 amino acid sequences with a cutoff of two and orphan VJ chains removed. In both cases, V gene identity was required and the CDR3 sequence similarity was evaluated across all of a cell pair's V(D)J chains.

**Statistics and reproducibility.** Spearman rank correlation test (two tailed) was performed for zonation analysis of the SMG serous cells across trachea, bronchi 2–3 and bronchi 4 locations for the three donors (A37, A41 and A42) with at least 20 cells in at least two locations. Correlation coefficients and *P* values were calculated per every donor separately. A Poisson linear mixed model was used for cell type composition analysis. Poisson regression with various metadata as covariates was applied to adjust confounding effects on the cell type count data as previously described[112,113]. We used location as a biological factor, and protocol, material (scRNA-seq versus snRNA-seq) and donor as technical factors in the model as random effects to overcome the collinearity (see Supplementary Notes in ref. [113] for more details).

Gene set enrichment analysis was done with g:GOSt method and g:SCS threshold with flat list in the gProfiler webpage https://biit.cs.ut.ee/gprofiler/gost. GOSt uses Fisher's one-tailed test, also known as cumulative hypergeometric probability.

Donors used for smFISH, IHC and Visium experiments are shown in Supplementary Table 9, smFISH and Visium experiments were performed once for each donor often across multiple sections, IHC staining was repeated at least twice depending on the markers used. Full staining figures, reproducibility and antibodies for protein staining from HPA can be queried online at https://www.proteinatlas.org/.

**Variance in gene expression.** To determine the effects of the metadata features on the expression data, a linear mixed model was used[39]. Genes expressed in less than 5% of the samples were filtered out. The count matrix was then normalized and log transformed. The percentages of variance in gene expression data explained by each metadata feature were obtained by fitting the linear mixed model. The Bayes factor was then computed to determine the gene-specific effects of some metadata features in the expression data, assigning an effect size and a local true sign rate (LTSR) for all genes analyzed. Genes presenting an LTSR value greater than 0.9 were considered substantially affected by the metadata feature analyzed. See Supplementary Notes in ref. [39] for more details.

**fGWAS analysis.** The fGWAS approach to determine disease-relevant cell types is described elsewhere[18]. Summary statistics for the selected GWAS study of Lung function (FEV1/FVC)[114] were obtained via Open Targets Genetics (https://genetics.opentargets.io/study/GCST007431; https://www.ebi.ac.uk/gwas/studies/GCST007431). The code used for fGWAS plots and for cell type proportion analysis is available here: https://github.com/natsuhiko/PHM.

**HLCA, COPD and IPF data analysis.** To assess PB-fibros in lung disease, we used both the HLCA extended[10] and Adams et al.[8] datasets. To look for differentially expressed genes between PB-fibros in COPD, fibroblast labels were transferred from the HLCA (which annotates PB-fibro based on our work) to Adams et al.[8] dataset. Differential genes were identified by Wilcoxon rank-sum test (*P* < 0.05). PB-fibros were rare, 74 in COPD and 20 in controls, but represented across 21 individuals (12 patients with COPD and 9 healthy controls). Genes displayed are a portion of the top-upregulated genes, which are implicated in Lung function (FEV1/FVC). The abundance of PB-fibros in disease was assessed in the HLCA and Adams et al.[8] datasets using PLMM as described above and MiloPy, which tests differential abundances on the KNN graph[115]. For PLMM analysis, covariates from the HLCA accounted for were 'sample', 'study', 'subject ID', 'sex', 'ethnicity', 'smoking status', 'condition' (disease, manually harmonized), 'subject type', 'sample type', 'sequencing platform', 'cells or nuclei' and 'anatomical region detailed unharmonized'. MiloPy analysis was performed using the standard workflow, the KNN graph was generated from the latent space already available in the extended HLCA (X_scanvi_emb) and 'study' was used as a covariate.

**COVID-19 data analysis.** To assess the impact of infection on IgA abundance, we used a dataset[71], with tracheal, bronchial and nasal epithelium brushings from children and adults. Although these samples differed from the deep airway biopsies taken in this study, they still contained some SMG epithelial cells and IgA plasma cells, showing a level of compatibility with the healthy samples used in our study. A total of 470 plasma cells from five donors (three healthy and two COVID+), each with at least 20 plasma cells, were pooled, clustered and classified into IgA, IgD, IgH and IgM isotypes based on expression levels of *IGHA1, IGHA2, IGHD, IGHG1, IGHG2, IGHG3, IGHG4* and *IGHM*. Proportions were calculated across cells for healthy and COVID+ donors separately.

**Cell–cell interaction analysis.** Cell–cell interaction from scRNA-seq data was predicted using CellChat[77]. B plasma subsets were combined (B-plasma) and cell types of interest (B-naive, B-memory, SMG duct, SMG mucous, SMG serous, CD4-naive/CD4-CM, CD4-EM/CD4-effector and CD4-TRM) were downsampled to 200 cells per cell type per donor. Analyses were performed both with individual CD4 T cell subsets and all CD4 subsets combined. Normalized count matrix along with cell annotation metadata was processed through the standard CellChat pipeline, except that the communication probability was calculated with a truncated mean of 10%.

**Reporting summary**

Further information on research design is available in the Nature Portfolio Reporting Summary linked to this article.

## Data availability

All transcriptomic datasets generated as part of the study are publicly available. The processed scRNA-seq, snRNA-seq and Visium ST data are available for browsing and download via our website www.lungcellatlas.org. The dataset (raw data and metadata) is available on the Human Cell Atlas Data Portal and on the European Nucleotide Archive (ENA) under accession number PRJEB52292 and BioStudies accession S-SUBS17. The Visium data are publicly available on ArrayExpress with the accession number E-MTAB-11640. Imaging data can be downloaded from European Bioinformatics Institute (EBI) BioImage Archive under accession number S-BIAD570. Additional data were accessed to support analysis and conclusions, which can be accessed through National Centre for Biotechnology Information Gene Expression Omnibus GSE136831, and GSE134174 and the HLCA integration, which can be accessed through github https://github.com/LungCellAtlas/HLCA.

## Code availability

The majority of the analysis was carried out using published and freely available software and code as stated in the Methods. Custom code

was used for Extended Data Figs. 1c and 8f and are available at https://github.com/elo073/5loclung/ (DOI: 10.5281/zenodo.7125810).

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

## Acknowledgements

We thank J. Eliasova for the graphical illustrations. L. Yang supported with a draft for graphical illustrations in BioRender. C. Dominguez Conde supported with a script for TCR clonotype sharing analysis. M. Prete's and Cellgen IT's computational support has been central to the analysis. J.E., L.Y., C.D.C, M.P. are affiliated with the Wellcome Sanger Institute. We are grateful to the organ donors, their families and the Collaborative Biorepository for Translational Medicine for the gift of human tissue. K.B.M. and S.A.T. are supported by Wellcome (WT211276/ Z/18/Z and Sanger core grant WT206194). E.M. is supported by ESPOD fellowship of EMBL-EBI and Sanger Institute. A.J.O. was supported by the European Respiratory Society and the European Union's H2020 research and innovation program under Marie Sklodowska-Curie grant agreement number 847462. K.T.M. is supported by an award from the Chan Zuckerberg Foundation. The project has received funding from the European Union's Horizon 2020 research and innovation program under grant agreement 874656. M.Z.N. acknowledges funding from an MRC Clinician Scientist Fellowship (MR/W00111X/1) and an MRC Rutherford Fellowship (MR/5005579/1). M.Z.N. and K.B.M. have been funded by the Rosetrees Trust (M944) and Action Medical Research (GN2911). K.B.W. acknowledges funding from University College London, Birkbeck MRC Doctoral Training Programme. This project has been made possible in part by grant 2019-202654 from the Chan Zuckerberg Foundation. This publication is part of the Human Cell Atlas (www.humancellatlas.org/publications).

## Author contributions

K.B.M., E.M. and A.J.O. conceived and designed the experiments; E.M., A.J.O., K.P., A.R.O., J.P.P., C.X., R.E., N.H. and R.G.H.L. carried out computational analysis; V.K. performed and optimized cell2location analysis; A.W-C. helped with experimental planning, sample management and spatial gene expression; L.M., L.B., A.K., E.P., A.H. and A.O. carried out tissue dissociation and sc and snRNA-seq experiments; M.D., L.T., S.P. and S.F.V. performed Visium ST and M.P. supervised RNAScope analysis; S.P. provided histology support. L.S.C provided pathology support. N.R., S.P. and A.J.O. carried out IHC and protein staining; K.M. and T.N. provided mouse tissue. P.H. and R.E. contributed to cell types annotation; K.M., N.G., K.S-P. provided human tissue samples; N.K. carried out statistical analysis and M.Y., K.B.W., R.G.H.L. and M.Z.N. shared unpublished data. O.A.B., M.C., O.S., S.A.T. and K.B.M provided funding, discussion and supervision and E.M., A.J.O., A.W-C., S.A.T. and K.B.M. wrote the manuscript.

## Competing interests

In the past three years, SAT has received remuneration for consulting and Scientific Advisory Board Membership from Genentech, Roche, Biogen, GlaxoSmithKline, Foresite Labs and Qiagen. SAT is a cofounder, board member and holds equity in Transition Bio. OS is a paid member of the Scientific Advisory Board of Insitro. The remaining authors declare no competing interests.

## Additional information

**Extended data** is available for this paper at https://doi.org/10.1038/s41588-022-01243-4.

**Correspondence and requests for materials** should be addressed to Sarah A. Teichmann or Kerstin B. Meyer.

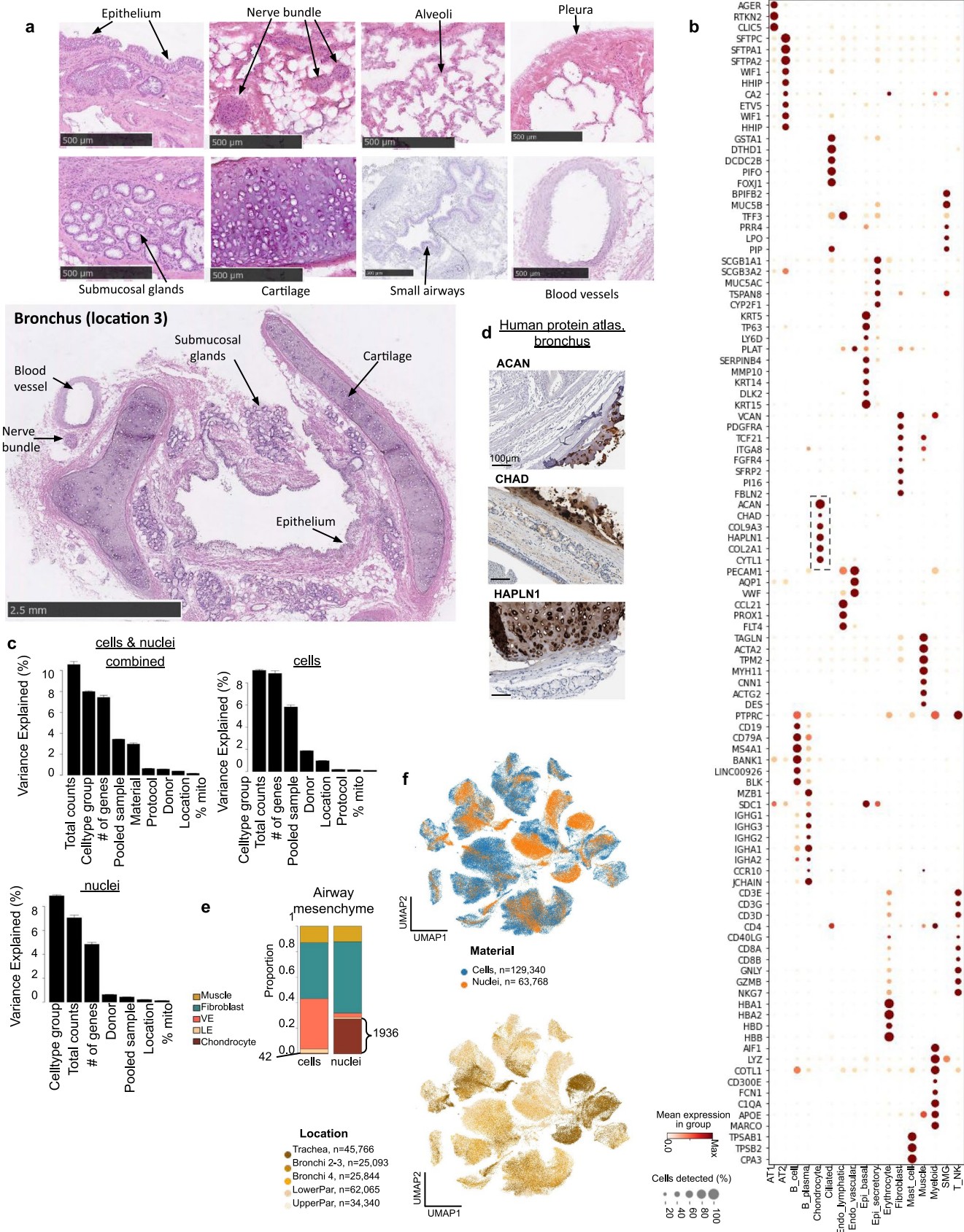

**Extended Data Fig. 1 | See next page for caption.**

**Extended Data Fig. 1 | Overview of human lung dataset across five locations.**
(**a**) H&E sections of full depth human tissue samples from multiple regions showing all major structures of the lungs and airways. (**b**) Expression of cell type marker genes in the master cell type groups, from both single cell and single nuclei RNA-seq combined. Color represents maximum normalised mean expression of marker genes in each cell group, and size indicates the proportion of cells expressing marker gene. Dashed box highlights chondrocyte marker genes. (**c**) Variance of gene expression explained by metadata variables in the combined sc/snRNA-seq dataset, scRNA-seq and snRNA-seq datasets. The whiskers correspond to 95% confidence intervals and the number of genes tested was 8,666 in cells/nuclei combined, 7,977 in cells, 7,260 in nuclei. 129,340 cells and 63,768 nuclei were analysed. (**d**) Protein staining of chondrocyte markers in the cartilage of human bronchus from the HPA. (**e**) Proportion of mesenchyme cell type groups in the airways from cells and nuclei. Numbers indicate chondrocytes in single cells versus single nuclei. (**f**) UMAP of sequencing material (cells or nuclei) and location (trachea, bronchi, parenchyma).

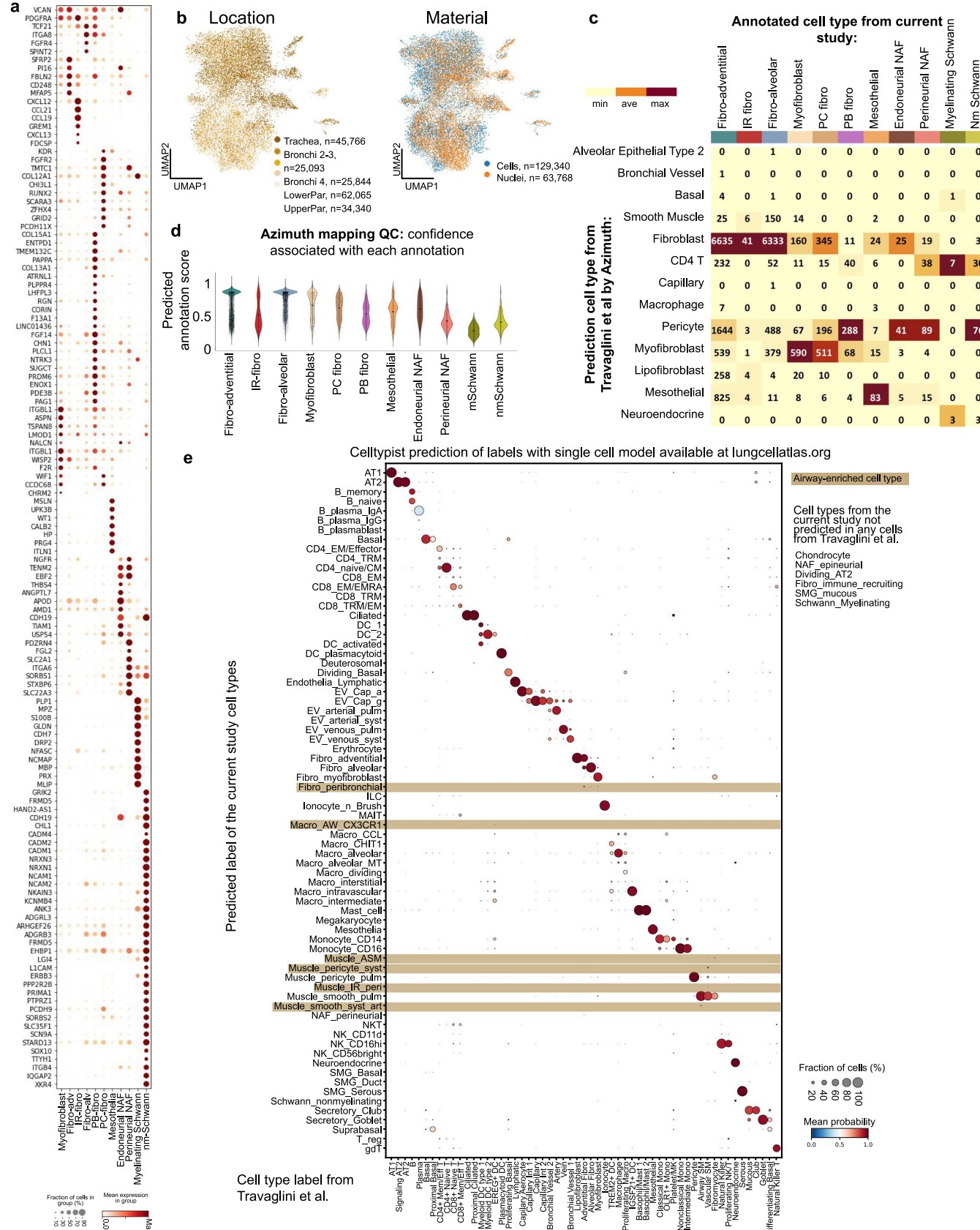

**Extended Data Fig. 2 | See next page for caption.**

**Extended Data Fig. 2 | Novel fibroblast subsets. (a)** Dot plot of marker gene expression for indicated cell types. (**b**) UMAP of location and sequencing material from fibroblasts. (**c**) Heatmap showing annotated cell types to the predicted labels for fibroblasts from Travaglini et al. (Travaglini et al. 2020) by the Azimuth tool, coloured by proportion. Labels by the proportion of annotated cells and the total number of cells mapping to the reference. (**d**) Violin plots with predicted annotation score for each of the annotated cell types to the reference. Small dots represent cells, circles represent mean values and bars show standard deviation. (**e**) Dot plot showing the cell type cross-validation by transferring cell type labels from our single cell dataset (row) to cell types from Travaglini et al. 2020 (column). For each column (each cell type from the Travaglini et al. 2020), size of a dot denotes the proportion of cells assigned to a given cell type in our dataset and colour denotes the average probability. Highlighting marks the airway-enriched cell types.

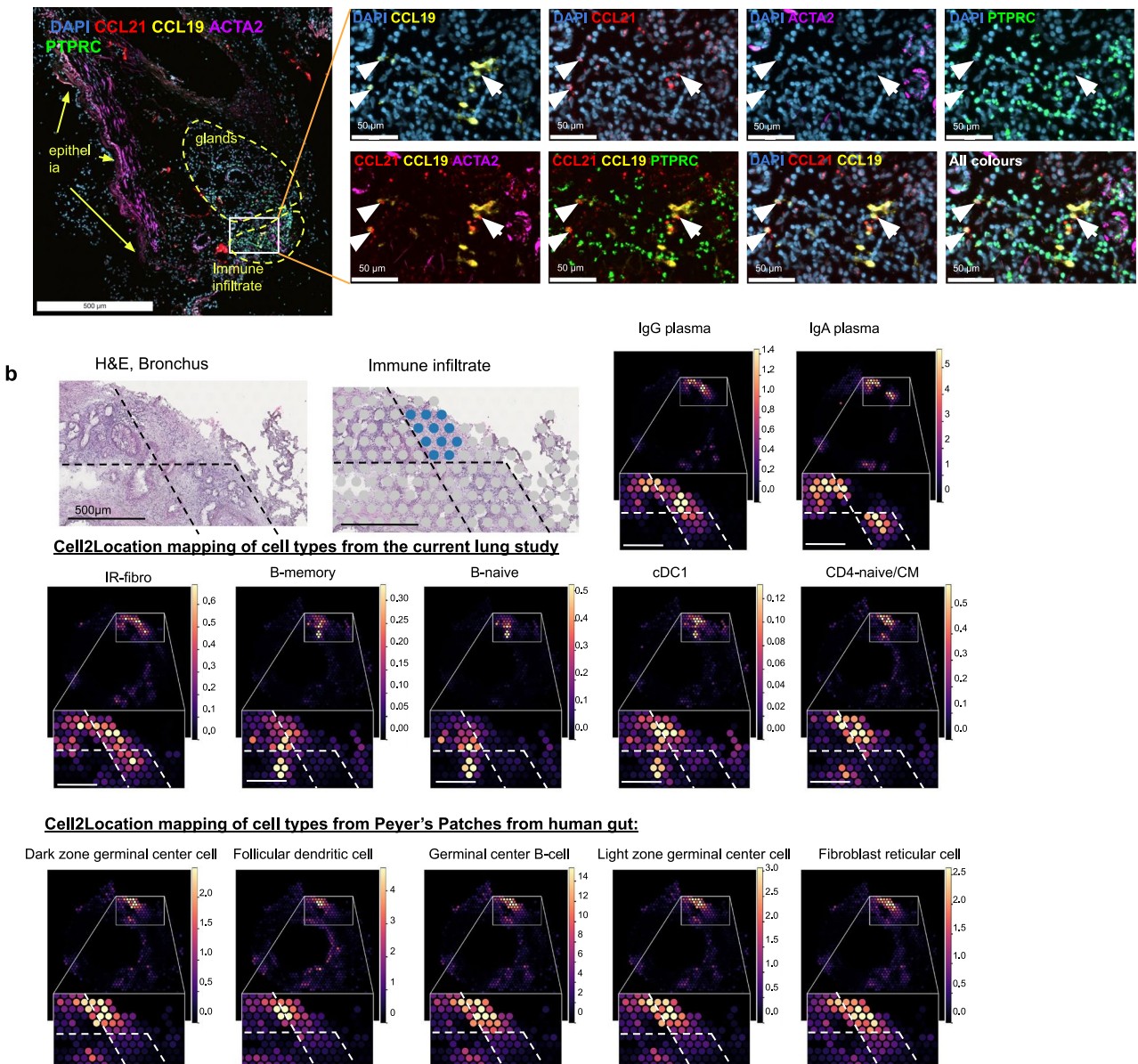

**Extended Data Fig. 3 | Validation of immune recruiting fibroblasts and their tissue localisation.** (**a**) smFISH staining in human bronchi tissue for IR-Fibro markers (CCL21, CCL19) showing independent localisation from immune cells (PTPRC) and smooth muscle cells (ACTA2) marked by arrows. (**b**) H&E staining on Visium ST with manually annotated immune infiltrate in blue. cell2location mapping density scores with zoom into the region of interest, showing density values for IR-Fibro and relevant immune cells from the current lung study as well as for germinal centre cell types from a gut[18]. Dashed lines are added for better visual comparison between the cell types and regions.

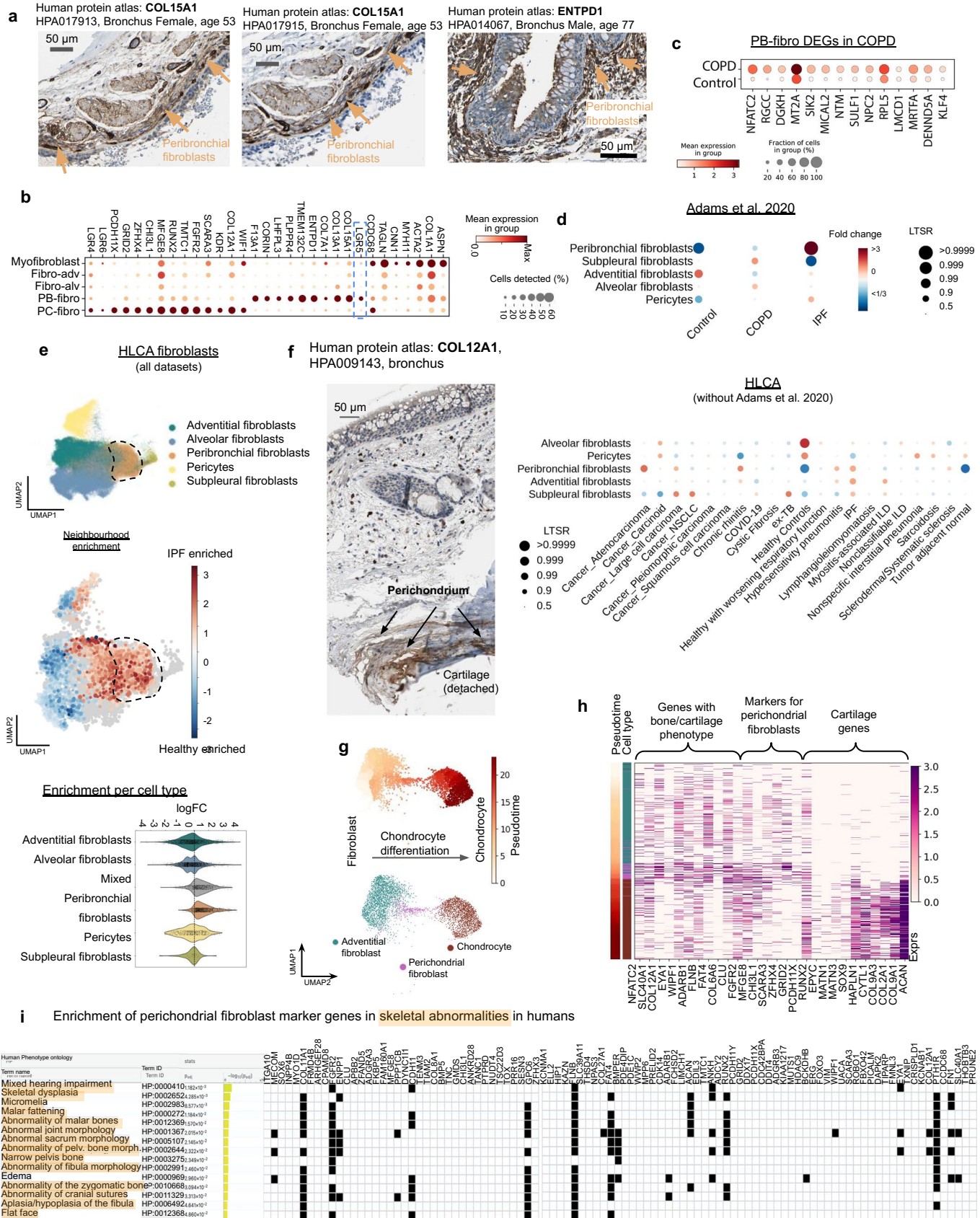

**Extended Data Fig. 4 | See next page for caption.**

**Extended Data Fig. 4 | Peribronchial and perichondrial fibroblasts. (a)** Protein staining of PB-Fibro markers (COL15A and ENTPD1) in human bronchus sections from the HPA. **(b)** Marker genes for PB-fibro and PC-fibro. **(c)** Upregulated genes in COPD patient's PB-fibro cells (74 cells, 12 donors) compared to controls (20 cells, 9 donors) from scRNAseq data9[8]. Selected upregulated genes associated with COPD or emphysema by GWAS (RGCC, DGKH, NTM, SULF1, NPC2, RPL5, LMCD1, MRTFA, DENND5A, KLF4) or in other studies (NFATC2, MT2A and SIK2). Wilcoxon rank sum test p < 0.05 (two sided), exact P values and full list is in Supplementary Table 10. **(d)** Cell type proportion analysis using PLMM to compare fibroblasts in the extended HLCA across disease conditions with fold changes and Local True Sign Rate (LTSR). Covariates are listed in the methods. Cell abundances were analysed in Adams et al. 2020 dataset only, and validated in the extended HLCA (minus Adams et al. 2020). **(e)** Milo cell type abundance analysis of fibroblasts from the HLCA comparing IPF patients and healthy controls. UMAP of fibroblast clusters, neighbourhood enrichment UMAP showing log fold change in IPF compared to healthy, and violin plot of log fold change of the neighbourhood for each cell, grouped by cell type. Dashed line highlights the region of PB fibros on the UMAPs. **(f)** Protein staining of PC-Fibro marker (COL12A1) in human bronchus from the HPA mapping to cartilage. **(g)** UMAP of adventitial fibroblasts, PC-fibro and chondrocytes from single nuclei data coloured by monocle 3 pseudotime and cell type. **(h)** Expression of genes associated with bone/cartilage function, markers of PC-fibro and cartilage genes in the nuclei as shown on **(g)**, ordered by pseudotime. **(i)** PC-fibro marker gene enrichment in Human Phenotype Ontology by g:Profiler.

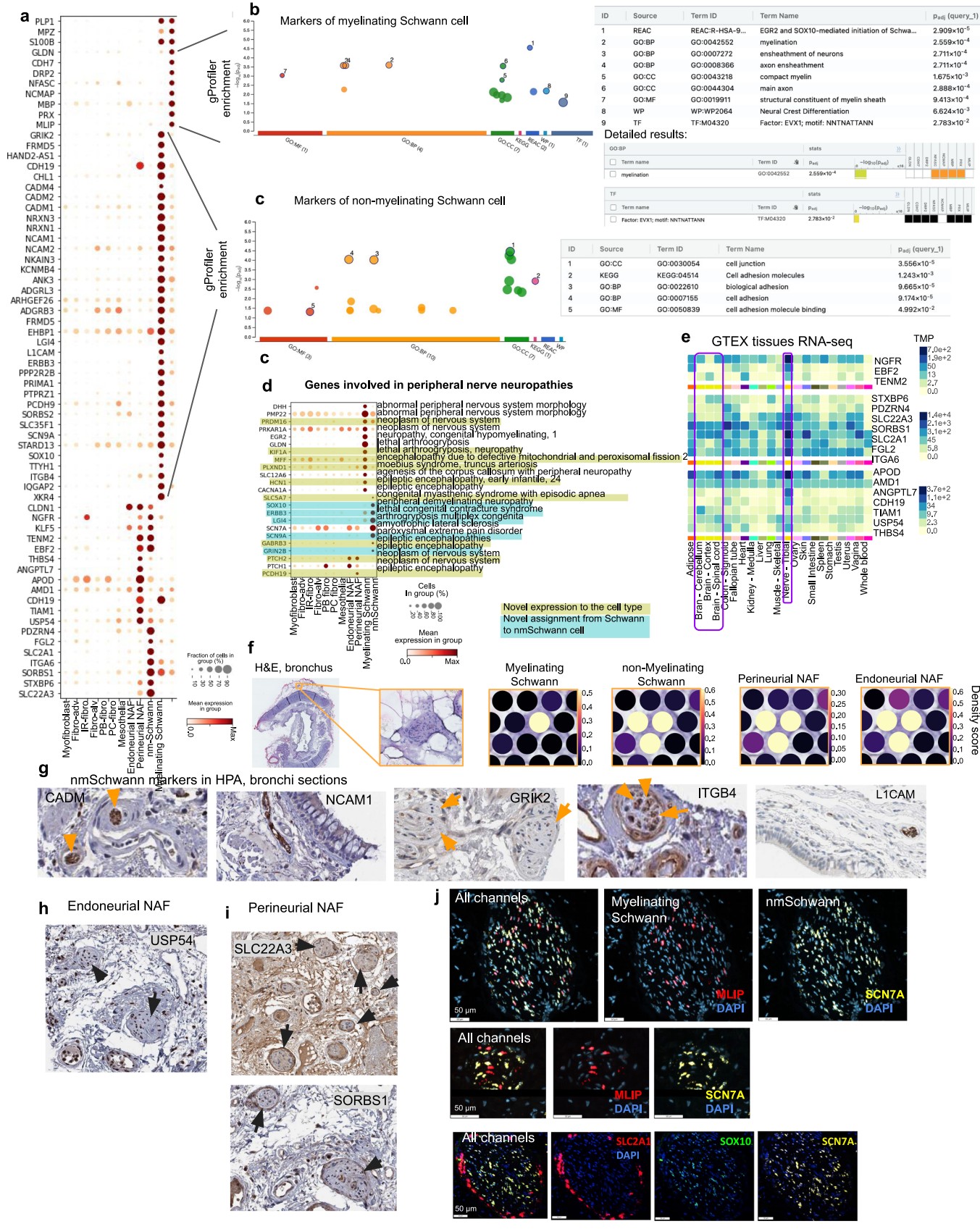

**Extended Data Fig. 5 | See next page for caption.**

**Extended Data Fig. 5 | Schwann cells and nerve-associated fibroblasts (NAF).** (**a**) Marker dot plot for myelinating, non-myelinating Schwann cells and for epi- and endoneurial NAF-s. (**b**, **c**) g:Profiler gene set enrichment results using g:GOSt method and g:SCS threshold and multiple testig correction with flat list as input for myelinating Schwann cell markers with detailed results for myelination and transcription factor EVX1 (**b**) and for non-myelinating Schwann cell markers (**c**). (**d**) Expression of neuropathy associated genes in Schwann and NAF cell types. Previously unknown cell type specific expression shown in colour: light green for novel expression pattern, light blue for distinguishing expression for nmSchwann cells. (**e**) Expression in Transcript per million (TPM) of NAF markers in GTEx bulk RNA-seq data. (**f**) Visium ST H&E staining of human bronchi, with zoom in on nerve bundle and cell2location cell type mapping density scores for Schwann and NAF cell types. (**g-i**) HPA antibody staining of (**g**) non-myelinating Schwann cell markers (CADM, GRIK2, NCAM1, ITGB4 and L1CAM) (**h**) endoneurial NAF marker (USP54) and (**i**) perineurial NAF markers (SLC22A3 and SORBS1) within the nerve bundles in human bronchus. Arrows indicate nerve bundles. (**j**) RNAscope staining for myelinating (MLIP) and non-myelinating (SCN7A, SOX10) Schwann cell and perineurial (SLC2A1) NAF specific genes in bronchial nerves.

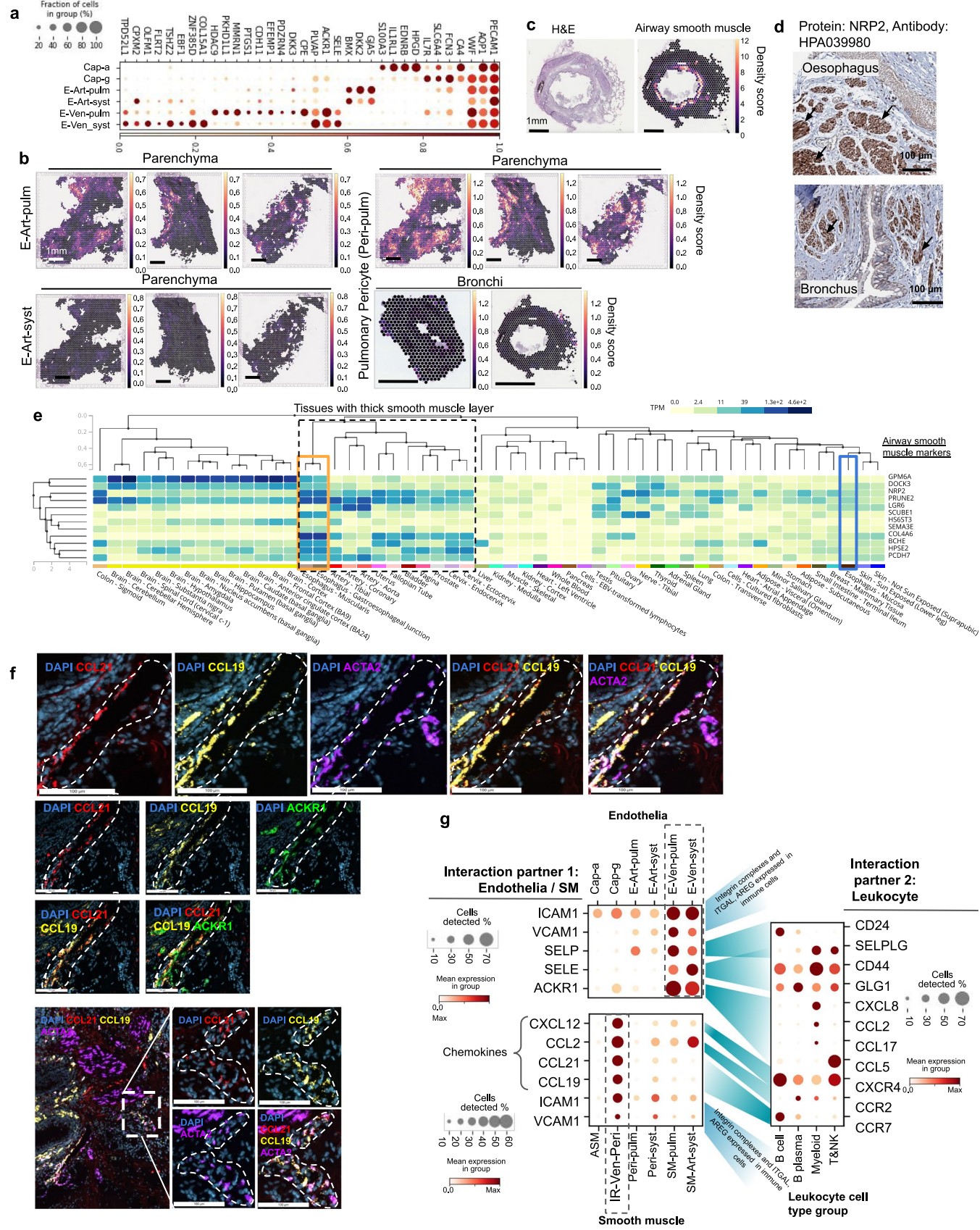

**Extended Data Fig. 6 | See next page for caption.**

**Extended Data Fig. 6 | Vascular and smooth muscle cell types. (a)** Markers dot plot for vascular endothelia. (**b**) cell2location density scores of pulmonary and vascular endothelium for parenchyma and bronchi Visium ST sections. (**c**) Bronchi section with H&E and cell2location analysis density score for airway smooth muscle population on a Visium ST slide. (**d**) NPR2 staining in oesophagus and bronchus from the HPA. Black arrows indicate the airway and oesophagus surrounding non-vascular smooth muscle. (**e**) ASM marker expression in all GTEx tissues. Tissues are ordered by unsupervised clustering based on expression similarity. The dotted line highlights tissues which are surrounded by a thick smooth muscle layer. The orange rectangle shows muscular tissues from oesophagus, and the blue rectangle shows the non-muscular mucous layer of oesophagus tissue. (**f**) IR-Ven-peri markers localise at the venous vessels in the airway. smFISH staining for IR-Ven-peri (CCL21, CCL19), venous endothelia (ACKR1) and smooth muscle (ACTA2) markers. (**g**) Leukocyte rolling and homing genes, and chemokines expressed in Endothelia and SM/Perivascular cells together with their interaction partners expression in immune cell groups. Interaction partners are indicated with blue shades.

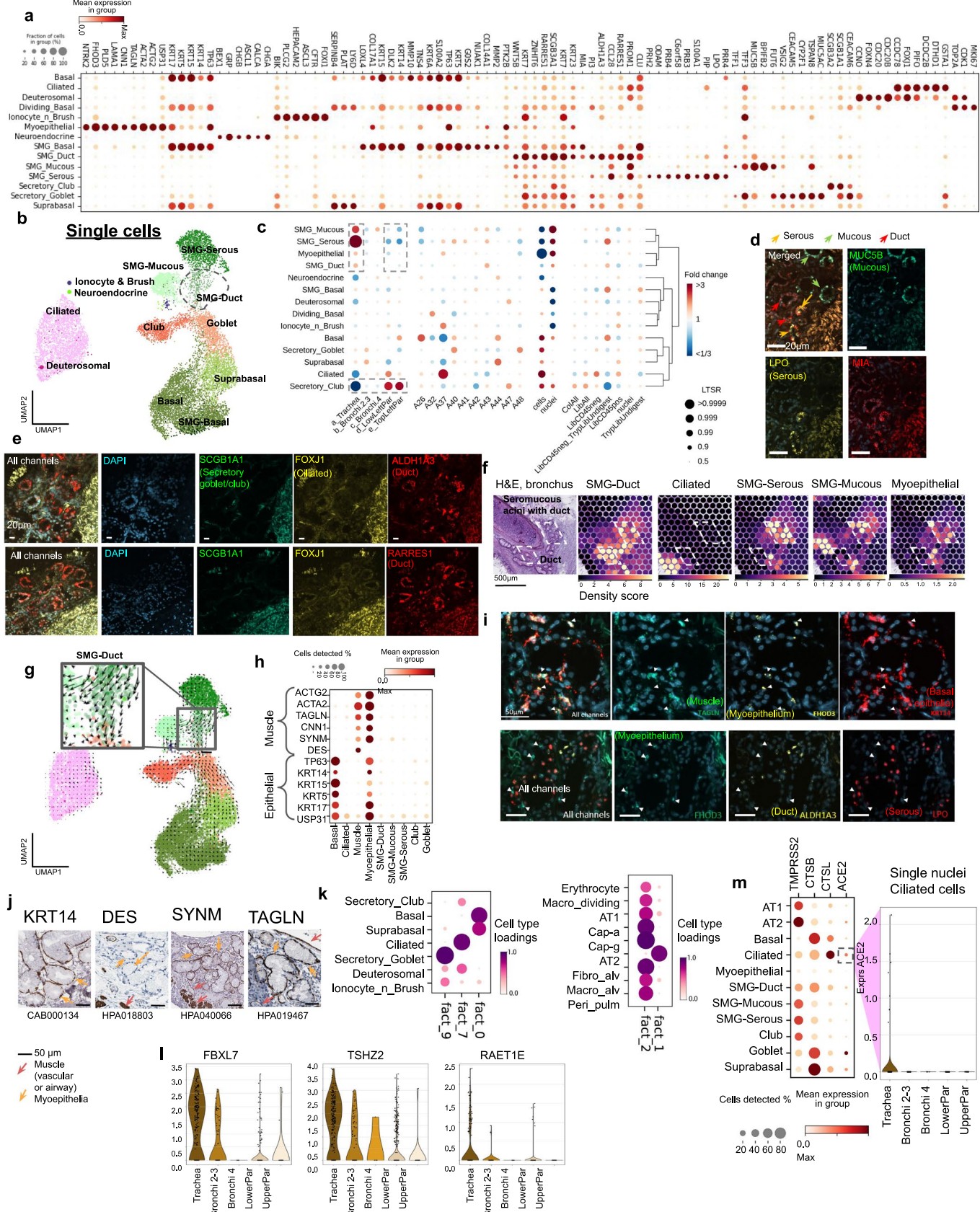

**Extended Data Fig. 7 | See next page for caption.**

**Extended Data Fig. 7 | Epithelial cell annotations and location specific ciliated cell gene expression.** (**a**) Marker gene expression dot plot for airway epithelial cells. (**b**) UMAP of airway epithelial cells from scRNA-seq data. (**c**) Cell type proportion analysis with fold changes and Local True Sign Rate (LTSR) score for all cell type groups with regards to variables shown. Cell numbers are in Supplementary Table 7. (**d**) smFISH staining for mucous (MUC5B), serous (LPO) and duct (MIA) cell markers in human bronchi sections. (**e**) smFISH staining of secretory goblet/club (SCGB1A1), ciliated (FOXJ1) and duct (ALDH1A3/RARRES1) in human bronchus section. (**f**) Visium ST H&E from bronchial section and cell2location density values for mapping duct, mucous, serous, ciliated and myoepithelial cells. (**g**) RNA velocity results on UMAP from scRNAseq of airway epithelia. Colours indicate cell types as in (**b**). (**h**) Myoepithelial marker gene expression dot plot. (**i**) smFISH staining for muscle (TAGLN), basal epithelia (KRT14), duct (ALDH1A3) and myoepithelium (FHOD3) in human bronchi sections. (**j**) HPA staining for muscle and epithelial marker proteins in human bronchial glands. (**k**) Unsupervised non-negative matrix factorisation (NMF) analysis of Visium ST cell2location results for 11 factors showing NMF factor loadings normalised per cell type (dot size and colour). Other factors/cell types are shown in Supplementary Fig. 4. (**l**) Violin plots of normalised log-transformed expression separated by location in the single nuclei RNA-seq data for 3 genes upregulated (methods) in nasopharyngeal carcinoma gene set from GSEA database with LTSR>0.9 consistently higher expressed in the trachea. (**m**) SARS-CoV-2 receptor and viral entry gene (ACE2) expression in ciliated cells from snRNA-seq data shown by location in a violin plot.

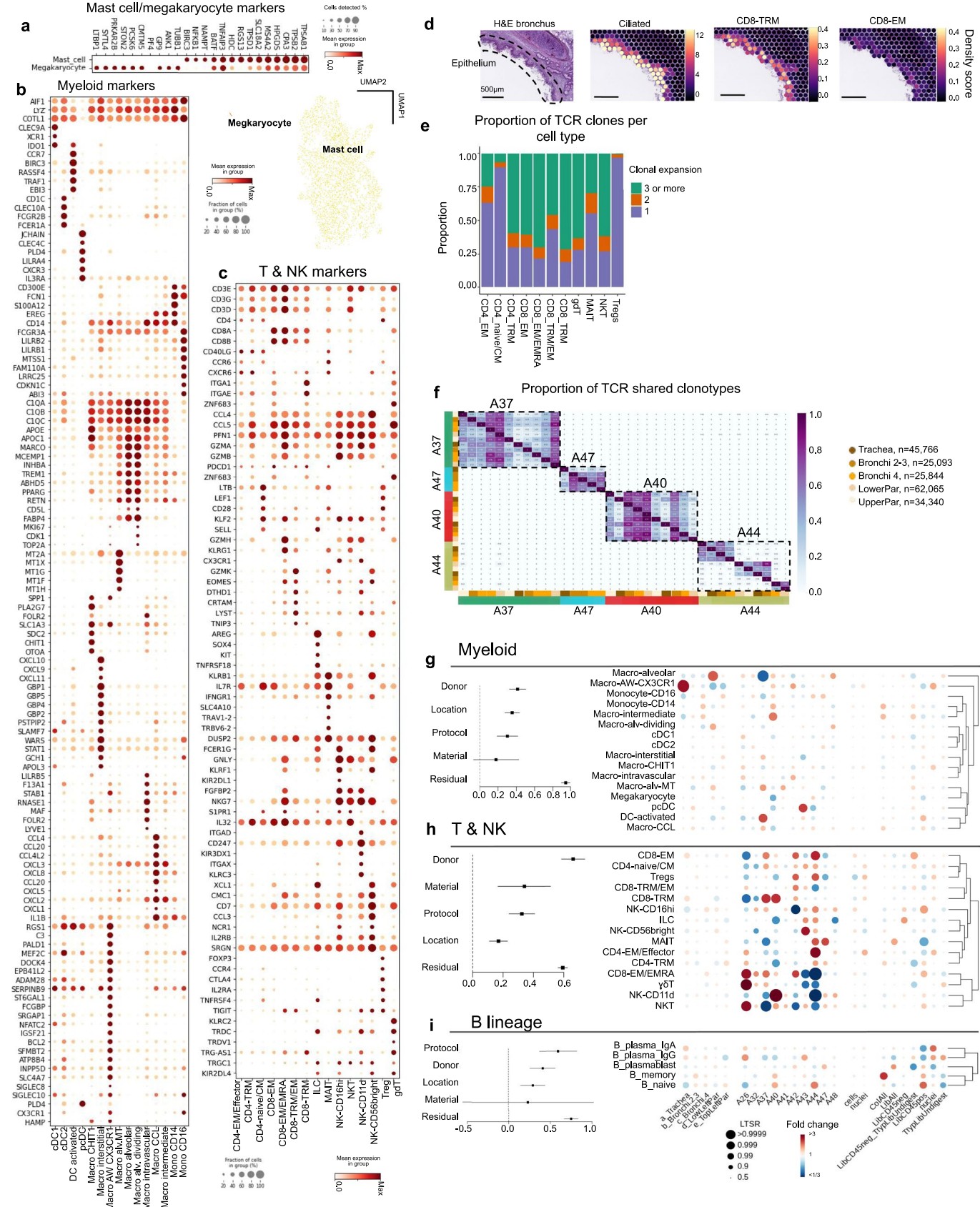

**Extended Data Fig. 8 | See next page for caption.**

**Extended Data Fig. 8 | Immune cell type groups.** (**a**) Marker gene expression plot along with UMAP for Megakaryocytes and Mast cells. (**b**) Marker genes dot plot for Myeloid cells. (**c**) Marker genes dot plot for T & NK cells. (**d**) cell2location density scores for CD8-TRM, Ciliated and CD8-EM cell types in human bronchi sections and corresponding H&E. (**e**) Fraction of clonally expanded cells in T & NKT cell types from VDJ data. (**f**) Proportion of shared TCR clonotypes between samples from VDJ data. Colour bars indicate location and donor. (**g-i**) Effects of location, donor, material and protocol on immune cell type proportions (assessed by PLMM) are shown by forest plots. Each square dot with an error bar shows the square root of variance explained by each factor and its 95% confidence interval, respectively. Dotplots show point estimates of fold changes and Local True Sign Rate (LTSR) for myeloid cell types (**g**), T & NK cell types (**h**) and B lineage cell types (**i**). The number of cells in each cell type group are in Supplementary Table 7.

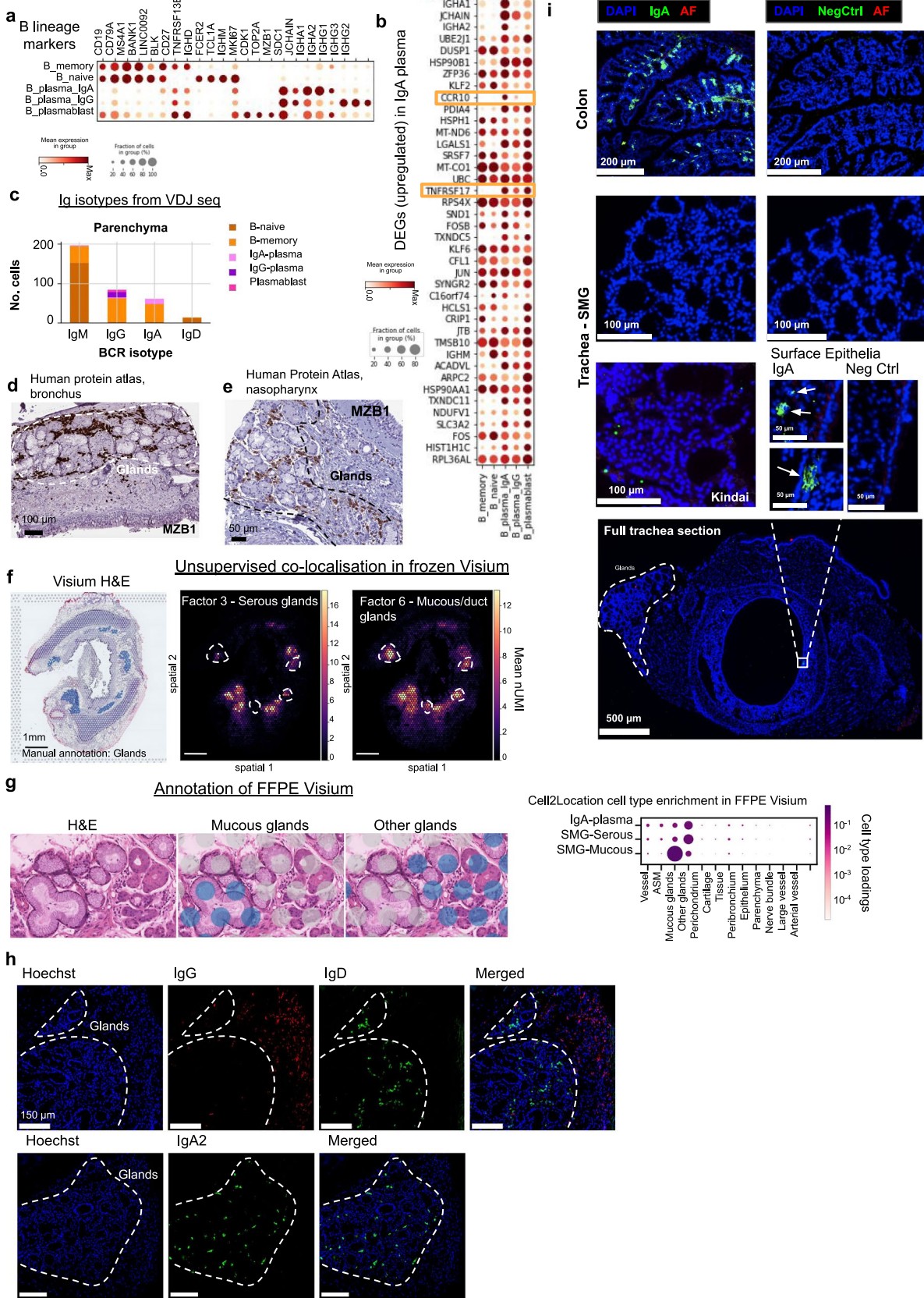

**Extended Data Fig. 9 | See next page for caption.**

**Extended Data Fig. 9 | Additional data on B lineage and IgA plasma cell localisation.** (**a**) Marker gene expression dot plot for B-lineage cells. (**b**) Gene expression dot plot for top differentially expressed genes between IgA and IgG plasma cells. (**c**) Number of B lineage cells with different Ig isotypes in parenchyma from the analysis of VDJ amplified libraries. (**d, e**) HPA staining for B plasma marker MZB1 in the bronchus.(**d**) and nasopharyngeal glands.(**e**). (**f**) Spatial localisation of selected NMF factors (total N=11, Figure 4f, Supplementary Fig. 4) from unsupervised NMF analysis of Visium ST cell2location results. The total cell abundance of constituent cell types for factors 3 and 6 are shown on a bronchi section, with H&E and manual gland annotations shown for reference. White dashed lines highlight mucous/duct (factor 6) specific areas. (**g**) An example of FFPE Visium slide region with mucous and non-mucous glands annotated per voxel, and enrichment of cell types by cell2location in the micro-anatomical tissue environments across 2 FFPE sections (trachea and bronchi 2-3) from 1 donor. (**h**) Multiplex IHC of human trachea for Ig isotypes (IgA, IgG, IgD) showing distinction between glands (dashed lines) and non-gland regions of tissue. (**i**) IgA staining in mouse colon and trachea from wild type C57/BL6 mice from Charles River, USA or Kindai University, Japan (where specified). Representative staining from 2 experiments, 3 sections per experiment. AF = autofluorescence (shown in red). Arrows showing clusters of IgA plasma cells at the epithelial surface in tracheal sections.

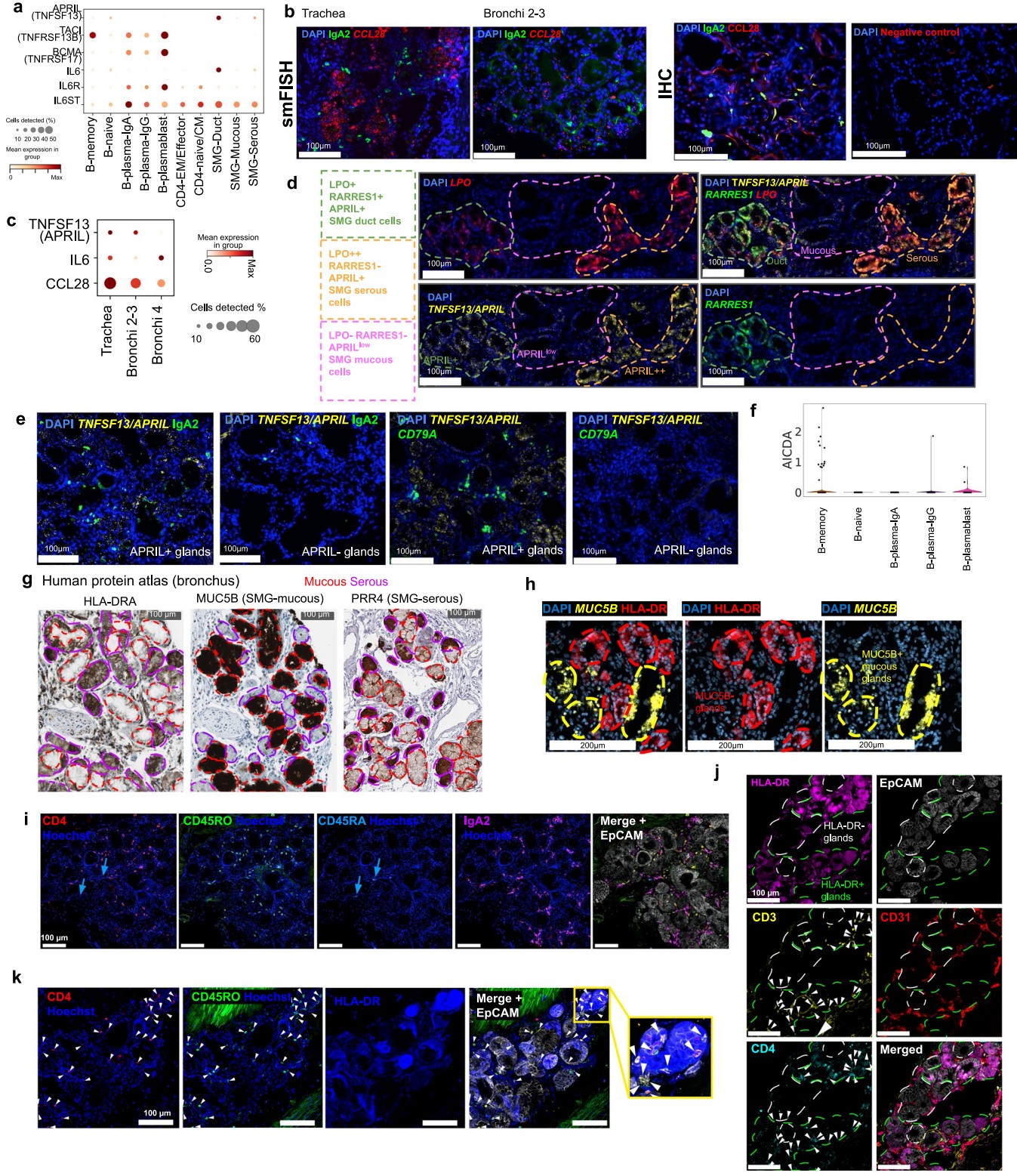

**Extended Data Fig. 10 | See next page for caption.**

**Extended Data Fig. 10 | Additional gland associated immune niche data - interactions of gland epithelial cells with immune cells.** (**a**) Expression dot plot of genes relevant for Figure 5h. (**b**) Co-staining of CCL28 RNA/protein and IgA2 protein in airway submucosal glands, left: smFISH; right: IHC for CCL28. (**c**) Expression of TNFSF13, IL-6 and CCL28 in SMG-Serous and Duct cell sc/snRNAseq data along the tracheobronchial tree. Change in CCL28 in serous cells was statistically significant with two-sided Spearman's rank correlation (methods) (p-values 1.6 × 10-8, 0.0034 and 1.3 × 10-5 for donors A37, A41 and A42 correspondingly). (**d**) smFISH on the human trachea with DAPI, TNFSF13/APRIL, RARRES1 and LPO. Dashed lines show regions of duct, mucous and serous glands. (**e**) smFISH (TNFSF13/APRIL, and CD79A) and IHC (IgA2) of human tracheal SMG showing APRIL+ and APRIL- glands annotated. (**f**) Violin plot for expression of AICDA in B cell subsets. (**g**) IHC from the HPA showing HLA-DR, MUC5B and PRR4 staining with HLA-DR+ regions corresponding with non-mucous areas of gland. Red/purple dashed lines indicate mucous/serous cells respectively based on morphology and/or marker gene expression. (**h**) smFISH (MUC5B) plus IHC (HLA-DR) staining in human airway SMG. (**i**) IHC staining of CD4, CD45RO (memory marker), CD45RA (naive marker), IgA2 and EpCAM in the SMG. Blue arrows indicate CD4+CD45RA+ cells. (**j**) IHC staining of HLA-DR, EpCAM, CD3, CD31 and CD4 in the airway SMG. Arrows point to CD3+CD4+ cells. (**k**) IHC staining of CD4, CD45RO, HLA-DR and EpCAM in the airway SMG showing close interactions between CD4+CD45RO+ T cells with HLA-DR+ glands. Arrows point to CD4+CD45RO+ cells.

# nature research

Sarah A Teichmann

# Reporting Summary

Nature Research wishes to improve the reproducibility of the work that we publish. This form provides structure for consistency and transparency in reporting. For further information on Nature Research policies, see our Editorial Policies and the Editorial Policy Checklist.

## Statistics

For all statistical analyses, confirm that the following items are present in the figure legend, table legend, main text, or Methods section.

| n/a | Confirmed | |
|---|---|---|
| ☐ | ☒ | The exact sample size (*n*) for each experimental group/condition, given as a discrete number and unit of measurement |
| ☐ | ☒ | A statement on whether measurements were taken from distinct samples or whether the same sample was measured repeatedly |
| ☐ | ☒ | The statistical test(s) used AND whether they are one- or two-sided<br>*Only common tests should be described solely by name; describe more complex techniques in the Methods section.* |
| ☐ | ☒ | A description of all covariates tested |
| ☐ | ☒ | A description of any assumptions or corrections, such as tests of normality and adjustment for multiple comparisons |
| ☒ | ☐ | A full description of the statistical parameters including central tendency (e.g. means) or other basic estimates (e.g. regression coefficient) AND variation (e.g. standard deviation) or associated estimates of uncertainty (e.g. confidence intervals) |
| ☐ | ☒ | For null hypothesis testing, the test statistic (e.g. *F*, *t*, *r*) with confidence intervals, effect sizes, degrees of freedom and *P* value noted<br>*Give P values as exact values whenever suitable.* |
| ☒ | ☐ | For Bayesian analysis, information on the choice of priors and Markov chain Monte Carlo settings |
| ☒ | ☐ | For hierarchical and complex designs, identification of the appropriate level for tests and full reporting of outcomes |
| ☐ | ☒ | Estimates of effect sizes (e.g. Cohen's *d*, Pearson's *r*), indicating how they were calculated |

*Our web collection on statistics for biologists contains articles on many of the points above.*

## Software and code

Policy information about availability of computer code

| Data collection | For this project sequencing data was generated on Illumina NovaSeq machines and analysed using the open source software listed in the next section. |
|---|---|
| Data analysis | The following software and code was used, with version number where available.<br><br>Space Ranger v1.1.0<br>Cell Ranger v4.0.0<br>Cell2location v0.1<br>SoupX v1.0.0<br>Harmony v1.0<br>BBKNN v1.4.1<br>scVI-tools v0.9.0<br>scanpy v1.7.1<br>gProfileR e106_eg53_p16_65fcd97<br>CellChat v1.1.1<br>scVelo package v0.2.1<br>Azimuth v0.4.1<br>CellTypist v1.2.0<br>Scirpy v0.6.0<br>Milopy v0.0.999<br>Omero v5.14.1<br>Imaris v9.7.0 |

Custom code is available on our Github repository https://github.com/elo073/5loclung, DOI 10.5281/zenodo.7125810.

For manuscripts utilizing custom algorithms or software that are central to the research but not yet described in published literature, software must be made available to editors and reviewers. We strongly encourage code deposition in a community repository (e.g. GitHub). See the Nature Research guidelines for submitting code & software for further information.

## Data

Policy information about availability of data

All manuscripts must include a data availability statement. This statement should provide the following information, where applicable:
- Accession codes, unique identifiers, or web links for publicly available datasets
- A list of figures that have associated raw data
- A description of any restrictions on data availability

All transcriptomic data generated as part of the study are publicly available. The processed scRNA-seq, snRNA-seq and Visium spatial transcriptomics data is available for browsing and download via our website www.lungcellatlas.org. The dataset (raw data and metadata) is available on the Human Cell Atlas Data Portal and on the European Nucleotide Archive (ENA) under accession number PRJEB52292 and BioStudies accession S-SUBS17. The Visium data is publicly available on ArrayExpress with the accession number E-MTAB-11640. Imaging data can be downloaded from European Bioinformatics Institute (EBI) BioImage Archive under accession number S-BIAD570. Additional data was accessed to support analysis and conclusions, which can be accessed through National Centre for Biotechnology Information Gene Expression Omnibus GSE136831, and GSE134174 and the Human Lung Cell Atlas integration which can be accessed through github https://github.com/LungCellAtlas/HLCA.

# Field-specific reporting

Please select the one below that is the best fit for your research. If you are not sure, read the appropriate sections before making your selection.

☒ Life sciences  ☐ Behavioural & social sciences  ☐ Ecological, evolutionary & environmental sciences

For a reference copy of the document with all sections, see nature.com/documents/nr-reporting-summary-flat.pdf

# Life sciences study design

All studies must disclose on these points even when the disclosure is negative.

| | |
|---|---|
| Sample size | Samples were obtained from a total of 12 human organ donors (7 donors for scRNA-seq, 7 donors for snRNA-seq and 6 donors for Visium ST as in Supplementary tables 1-3).<br>Total number of single cell and single nuclei transcriptomes analysed: 193,108<br>No Sample size calculation was carried out as we are not comparing distinct phenotypes but describing the cell types present in normal healthy lung and airway. Due to the rarity of organ lung donor availability, samples were profiled in depth using multiple techniques rather than including many donors. The sample size in our study is sufficient as it is more than most other published single cell studies that have profiled healthy lung. All described populations were identified in at least three individuals. |
| Data exclusions | Thresholds common in single cell studies were applied to exclude poor quality cells (see methods for details). No individuals were excluded from the analysis. Five sections from the fresh frozen Visum ST experiment were excluded from Cell2location calculations due to poor quality. |
| Replication | All findings we report are supported by robust statistical analysis that is outlined in detail in the methods section of the paper. Any cell types we report were found in multiple individuals. For spatial transcriptomics analysis, a total of 20 sections were profiled however 5 were excluded due to low quality as stated in the methods. Cell type mapping by Visium spatial transcriptomics was reproducible in at least 3 sections.<br><br>When validating cell types by RNAscope, a minimum of 2 sections from 2 donors were stained and representative sections shown. For multiplex IHC experiments, 3 donor airway samples (both bronchi and trachea) were stained with at least 1 section per location per donor and representative sections shown. All replicate experiments were successful and reflect the results reported in the study. |
| Randomization | Randomisation was not applicable in the study. Patients that were available within organ donor programme were analysed immediately as fresh samples. The patients for frozen samples were pooled and analysed as soon as 5 patients were available. Sections for spatial methods were used specifically after screening for relevant structures on the sections. |
| Blinding | Blinding was not applicable to this study because all of our donors were healthy lung organ donors. |

# Reporting for specific materials, systems and methods

We require information from authors about some types of materials, experimental systems and methods used in many studies. Here, indicate whether each material, system or method listed is relevant to your study. If you are not sure if a list item applies to your research, read the appropriate section before selecting a response.

## Materials & experimental systems

| n/a | Involved in the study |
|-----|----------------------|
| ☐ | ☒ Antibodies |
| ☒ | ☐ Eukaryotic cell lines |
| ☒ | ☐ Palaeontology and archaeology |
| ☐ | ☒ Animals and other organisms |
| ☐ | ☒ Human research participants |
| ☒ | ☐ Clinical data |
| ☒ | ☐ Dual use research of concern |

## Methods

| n/a | Involved in the study |
|-----|----------------------|
| ☒ | ☐ ChIP-seq |
| ☒ | ☐ Flow cytometry |
| ☒ | ☐ MRI-based neuroimaging |

# Antibodies

| | |
|---|---|
| Antibodies used | Antibody details (supplier, catalog number, conjugate, clone name, cycle position for multiplexing and dilution) are given in Supplementary Table 6. |
| Validation | All antibodies are commercially available and validated, and we provide links to available technical datasheet below:<br>Hoechst 33258: https://biotium.com/wp-content/uploads/2016/12/PI-40044-40045.pdf<br>CD3: https://www.biolegend.com/fr-ch/products/alexa-fluor-488-anti-human-cd3-antibody-2726?pdf=true&displayInline=true&leftRightMargin=15&topBottomMargin=15&filename=Alexa%20Fluor%C2%AE%20488%20anti-human%20CD3%20Antibody.pdf<br>CD31: https://www.thermofisher.com/order/genome-database/dataSheetPdf?producttype=antibody&productsubtype=antibody_primary&productId=12-0319-42&version=224<br>HLA-DR: https://www.abcam.com/alexa-fluor-647-hla-dr-antibody-tal-1b5-ab223907.pdf<br>IgD: https://www.biolegend.com/en-us/products/alexa-fluor-488-anti-human-igd-antibody-7758?pdf=true&displayInline=true&leftRightMargin=15&topBottomMargin=15&filename=Alexa%20Fluor%C2%AE%20488%20anti-human%20IgD%20Antibody.pdf<br>CD4: https://www.biolegend.com/it-it/products/alexa-fluor-647-anti-human-cd4-antibody-2728?pdf=true&displayInline=true&leftRightMargin=15&topBottomMargin=15&filename=Alexa%20Fluor%C2%AE%20647%20anti-human%20CD4%20Antibody.pdf<br>IgA2: https://resources.southernbiotech.com/techbul/9140.pdf<br>EpCAM: https://www.thermofisher.com/order/genome-database/dataSheetPdf?producttype=antibody&productsubtype=antibody_primary&productId=50-9326-42&version=224<br>Phalloidin: https://www.biolegend.com/en-us/products/flash-phalloidin-green-488-13950?pdf=true&displayInline=true&leftRightMargin=15&topBottomMargin=15&filename=Flash%20Phalloidin%E2%84%A2%20Green%20488.pdf<br>CD45: https://www.biolegend.com/en-us/products/apc-anti-human-cd45-antibody-705?pdf=true&displayInline=true&leftRightMargin=15&topBottomMargin=15&filename=APC%20anti-human%20CD45%20Antibody.pdf<br>CD45RO: https://www.biolegend.com/en-us/products/alexa-fluor-488-anti-human-cd45ro-antibody-3340?pdf=true&displayInline=true&leftRightMargin=15&topBottomMargin=15&filename=Alexa%20Fluor%C2%AE%20488%20anti-human%20CD45RO%20Antibody.pdf<br>CD45RA:<br>https://www.biolegend.com/en-us/products/pe-anti-human-cd45ra-antibody-687?pdf=true&displayInline=true&leftRightMargin=15&topBottomMargin=15&filename=PE%20anti-human%20CD45RA%20Antibody.pdf<br>CCL28: https://www.atlasantibodies.com/api/print_datasheet/HPA077434.pdf<br>anti-rabbit IgG AF647: https://www.abcam.com/goat-rabbit-igg-hl-alexa-fluor-647-ab150079.html (link to datasheet download, antibody has 189 references as of 12 September 2022)<br>anti-mouse IgA PE: https://www.thermofisher.com/order/genome-database/dataSheetPdf?producttype=antibody&productsubtype=antibody_primary&productId=12-5994-81&version=251<br>Streptavidin PE: https://www.biolegend.com/it-it/products/pe-streptavidin-1475?pdf=true&displayInline=true&leftRightMargin=15&topBottomMargin=15&filename=PE%20Streptavidin.pdf&v=20220609101609 |

# Animals and other organisms

Policy information about studies involving animals; ARRIVE guidelines recommended for reporting animal research

| | |
|---|---|
| Laboratory animals | Wild-type C57/BL6 mouse samples were obtained from Kindai University, Japan (courtesy of Prof. Takashi Nakayama) and Charles River, USA (AMSbio). Male (colon samples) and female (all other samples) mice were and used at 8–10 weeks old. Mice from Kindai University were housed in specific pathogen free conditions with 12 hour light/dark cycle (light on from 7am to 7pm) at ambient temperature 22±1°C and 55±10% humidity. All animal experiments for mice obtained from Kindai University were approved by the Centre of Animal Experiments at Kindai University. Mice from Charles River were housed in specific pathogen free conditions with 12 hour light/dark cycle at ambient temperature 21±2°C and 30-70% humidity. Mouse tissue from Charles River was purchased from a certified animal supplier through AMSbio, with an internal ethical approval process for broadly defined research use. |
| Wild animals | No wild animals were used. |
| Field-collected samples | No field-collected samples were used. |
| Ethics oversight | *Identify the organization(s) that approved or provided guidance on the study protocol, OR state that no ethical approval or guidance* |

Ethics oversight | *was required and explain why not.*

Note that full information on the approval of the study protocol must also be provided in the manuscript.

# Human research participants

Policy information about <u>studies involving human research participants</u>

Population characteristics | All participants were deceased organ donors without prior history of lung disease. Patient characteristics such as age, BMI and sex are given in the Supplementary Table 1.

Recruitment | Samples from deceased organ donors were provided by the Cambridge Biorepository for Translational Medicine, and individual donors were selected on the basis of organ availability.

Ethics oversight | Samples were obtained from deceased transplant organ donors by the Cambridge Biorepository for Translational Medicine (CBTM) with informed consent from the donor families and approval from the NRES Committee of East of England – Cambridge South (15/EE/0152). This consent includes generation of open-access genetic sequencing data and publication in open access journals in line with Wellcome Trust policy. CBTM operates in accordance with UK Human Tissue Authority guidelines.

Note that full information on the approval of the study protocol must also be provided in the manuscript.

