## [Peer Review File · Nature Genetics]

Peer Review Information

Manuscript Title: A spatially resolved atlas of the human lung characterizes a gland-associated immune niche

Corresponding author name(s): Dr Kerstin Meyer

Reviewer Comments & Decisions:

Decision Letter, initial version:
--

5th Aug 2022

Dear Dr. Meyer,

Thank you for submitting your revised manuscript "A spatial multi-omics atlas of the human lung reveals a novel gland-associated immune niche" (NG-A60673-T). It has now been seen by the original referees and their comments are below. The reviewers find that the paper has improved in revision, and therefore we'll be happy in principle to publish it in Nature Genetics, pending minor revisions to comply with our editorial and formatting guidelines.

I think this is the fastest accept-in-principle decision I've ever been able to give an author!

I'm very pleased at this happy outcome!

Sincerely,

Safia Danovi
Editor
Nature Genetics

Reviewer #1 (Remarks to the Author):

The authors have satisfactorily addressed all of my comments.

Reviewer #2 (Remarks to the Author):

This paper represents an analysis of unique human lung environments via a multi-omics approach. The edits to the paper in response to the original reviewer comments do clarify the comparisons and provide a more nuanced discussion. It is appreciated that the murine studies did not identify an IgA SMG niche which certainly limits the functional analyses possible. While the paper remains largely descriptive, it does annotate some interesting niches of the human lung, which as pointed out by the authors are likely to be a springboard for the higher resolution studies in the future. It also has better discussion related to disease relevance.

Reviewer #3 (Remarks to the Author):

I'm comfortable with the response of the authors to original critique. I think they address all of my concerns. I'm fully supportive of publication of this paper.

Author Rebuttal to Initial comments

None

Final Decision Letter:

25th Oct 2022

Dear Dr. Meyer,

I am delighted to say that your manuscript "A spatially resolved atlas of the human lung characterizes a gland-associated immune niche" has been accepted for publication in an upcoming issue of Nature Genetics.

Your paper will be published online after we receive your corrections and will appear in print in the next available issue. You can find out your date of online publication by contacting the Nature Press Office (press@nature.com) after sending your e-proof corrections. Now is the time to inform your Public Relations or Press Office about your paper, as they might be interested in promoting its publication. This will allow them time to prepare an accurate and satisfactory press release. Include your manuscript tracking number (NG-A60673R) and the name of the journal, which they will need when they contact our Press Office.

Please note that *Nature Genetics* is a Transformative Journal (TJ). Authors may publish their research with us through the traditional subscription access route or make their paper immediately open access through payment of an article-processing charge (APC). Authors will not be required to make a final decision about access to their article until it has been accepted. [Find out more about Transformative Journals](https://www.springernature.com/gp/open-research/transformative-journals)

Authors may need to take specific actions to achieve [compliance with funder and institutional open access mandates](https://www.springernature.com/gp/open-research/funding/policy-compliance-faqs). If your research is supported by a funder that requires immediate open access (e.g. according to [Plan S principles](https://www.springernature.com/gp/open-research/plan-s-compliance)) then you should select the gold OA route, and we will direct you to the compliant route where possible. For authors selecting the subscription publication route, the journal's standard licensing terms will need to be accepted, including [self-archiving-and-license-to-publish](https://www.nature.com/nature-portfolio/editorial-policies/self-archiving-and-license-to-publish). Those licensing terms will supersede any other terms that the author or any third party may assert apply to any version of the manuscript.

Please note that Nature Portfolio offers an immediate open access option only for papers that were first submitted after 1 January, 2021.

If you have not already done so, we invite you to upload the step-by-step protocols used in this manuscript to the Protocols Exchange, part of our on-line web resource, natureprotocols.com. If you complete the upload by the time you receive your manuscript proofs, we can insert links in your article that lead directly to the protocol details. Your protocol will be made freely available upon publication of your paper. By participating in natureprotocols.com, you are enabling researchers to more readily reproduce or adapt the methodology you use. [Natureprotocols.com](http://natureprotocols.com) is fully searchable, providing your protocols and paper with increased utility and visibility. Please submit your protocol to <https://protocolexchange.researchsquare.com/>. After entering your [nature.com](http://www.nature.com) username and password you will need to enter your manuscript number (NG-A60673R). Further information can be found at <https://www.nature.com/nature-portfolio/editorial-policies/reporting-standards#protocols>

I'm absolutely delighted to be publishing this paper. It's been a pleasure working with you and I hope we'll get another chance to do so in the future.

Sincerely,

Safia

Safia Danovi
Editor
Nature Genetics

Click here if you would like to recommend Nature Genetics to your librarian

<http://www.nature.com/subscriptions/recommend.html#forms>

** Visit the Springer Nature Editorial and Publishing website at http://editorial-jobs.springernature.com?utm_source=ejp_NGen_email&utm_medium=ejp_NGen_email&utm_campaign=ejp_NGen for more information about our career opportunities. If you have any questions please click [here](mailto:editorial.publishing.jobs@springernature.com). **